# No Pose, No Problem: Surprisingly Simple 3D Gaussian Splats from Sparse Unposed Images

**Botao Ye**[1,2]      **Sifei Liu**[3]      **Haofei Xu**[1]      **Xueting Li**[3]

**Marc Pollefeys**[1,4]      **Ming-Hsuan Yang**[5]      **Songyou Peng**[1*]

[1]ETH Zurich    [2]ETH AI Center    [3]NVIDIA    [4]Microsoft    [5]UC Merced

https://noposplat.github.io

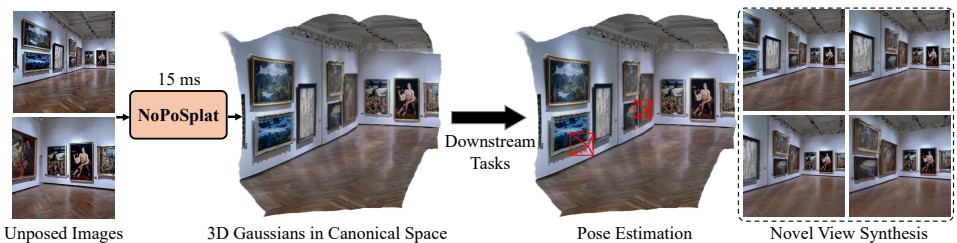

Figure 1: **NoPoSplat**. Given sparse *unposed* images, our method reconstructs 3D Gaussians of different views in a canonical space using a feed-forward network. The resulting 3D Gaussians can be utilized for accurate relative camera pose estimation and high-quality novel view synthesis. The input images for illustration are extracted from a Sora-generated video.

## Abstract

We introduce NoPoSplat, a feed-forward model capable of reconstructing 3D scenes parameterized by 3D Gaussians from *unposed* sparse multi-view images. Our model, trained exclusively with photometric loss, achieves real-time 3D Gaussian reconstruction during inference. To eliminate the need for accurate pose input during reconstruction, we anchor one input view's local camera coordinates as the canonical space and train the network to predict Gaussian primitives for all views within this space. This approach obviates the need to transform Gaussian primitives from local coordinates into a global coordinate system, thus avoiding errors associated with per-frame Gaussians and pose estimation. To resolve scale ambiguity, we design and compare various intrinsic embedding methods, ultimately opting to convert camera intrinsics into a token embedding and concatenate it with image tokens as input to the model, enabling accurate scene scale prediction. We utilize the reconstructed 3D Gaussians for novel view synthesis and pose estimation tasks and propose a two-stage coarse-to-fine pipeline for accurate pose estimation. Experimental results demonstrate that our pose-free approach can achieve superior novel view synthesis quality compared to pose-required methods, particularly in scenarios with limited input image overlap. For pose estimation, our method, trained without ground truth depth or explicit matching loss, significantly outperforms the state-of-the-art methods with substantial improvements. This work makes significant advances in pose-free generalizable 3D reconstruction and demonstrates its applicability to real-world scenarios. Code and trained models are available on our project page.

## 1 Introduction

We address the problem of reconstructing a 3D scene parameterized by 3D Gaussians from *unposed* sparse-view images (as few as two) using a feed-forward network. While current SOTA generalizable 3D reconstruction methods (Charatan et al., 2024; Chen et al., 2024), which aim to predict 3D radiance fields using feed-forward networks, can achieve photorealistic results without per-scene optimization, they require accurate camera poses of input views as input to the network. These

---

*Songyou Peng is currently at Google DeepMind, with this work mainly done at ETH Zurich.

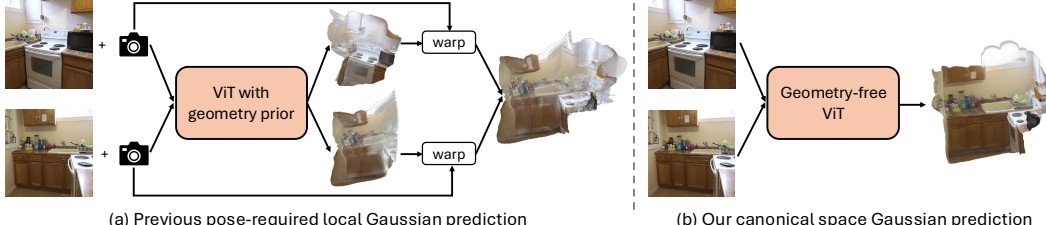

(a) Previous pose-required local Gaussian prediction        (b) Our canonical space Gaussian prediction

Figure 2: **Comparison with pose-required sparse-view 3D Gaussian splatting pipeline.** Previous methods first generate Gaussians in each local camera coordinate system and then transform them into a world coordinate system using camera poses. In contrast, NoPoSplat directly outputs 3D Gaussians of all views in a *canonical* space, facilitating a more coherent fusion of multi-view geometric content (see Sec. 3.3 for details). Furthermore, our backbone does not incorporate geometry priors that rely on substantial image overlap (such as epipolar geometry in pixelSplat (Charatan et al., 2024) or cost volume in MVSplat (Chen et al., 2024)). Consequently, NoPoSplat demonstrates better geometric detail preservation when the overlap between input views is limited.

poses are typically obtained from dense videos using structure-from-motion (SfM) methods, such as COLMAP (Schonberger & Frahm, 2016). This requirement is impractical for real-world applications, as these methods necessitate poses from dense videos even if only two frames are used for 3D reconstruction. Furthermore, relying on off-the-shelf pose estimation methods increases inference time and can fail in textureless areas or images without sufficient overlap.

Recent methods (Chen & Lee, 2023; Smith et al., 2023; Hong et al., 2024a) aim to address this challenge by integrating pose estimation and 3D scene reconstruction into a single pipeline. However, the quality of novel view renderings from these methods lags behind SOTA approaches that rely on known camera poses (Chen et al., 2024). The performance gap stems from their sequential process of alternating between pose estimation and scene reconstruction. Errors in pose estimation degrade the reconstruction, which in turn leads to further inaccuracies in pose estimation, creating a compounding effect. In this work, we demonstrate that reconstructing the scene entirely without relying on camera poses is feasible, thereby eliminating the need for pose estimation. We achieve this by directly predicting scene Gaussians in a canonical space, inspired by the success of the recent 3D point cloud reconstruction methods (Wang et al., 2024b; Leroy et al., 2024). However, unlike DUSt3R, we show that the generalizable reconstruction network can be trained with photometric loss only without ground truth depth information and thus can leverage more widely available video data (Zhou et al., 2018; Liu et al., 2021; Ling et al., 2024).

Specifically, as shown in Fig. 2 (b), we anchor the local camera coordinate of the first view as the canonical space and predict the Gaussian primitives (Kerbl et al., 2023) for all input views within this space. Consequently, the output Gaussians will be aligned to this canonical space. This contrasts with previous methods (Charatan et al., 2024; Chen et al., 2024), as illustrated in Fig. 2 (a), where Gaussian primitives are first predicted in each local coordinate system and then transformed to the world coordinate using the camera pose and fused together. Compared to the transform-then-fuse pipeline, we require the network to learn the fusion of different views directly within the canonical space, thereby eliminating misalignments introduced by explicit transformations (see Fig. 5).

Although the proposed pose-free pipeline is simple and promising, we observe significant scale misalignment in the rendered novel views compared to the ground truth (see Fig. 8), *i.e.*, the scene scale ambiguity issue. Upon analyzing the image projection process, we find that the camera's focal length is critical to resolving this scale ambiguity. This is because the model reconstructs the scene solely based on the image appearance, which is influenced by the focal length. Without incorporating it, the model struggles to recover the scene at the correct scale. To address this issue, we design and compare three different methods for embedding camera intrinsics and find that simply converting the intrinsic parameters into a feature token and concatenating it with the input image tokens enables the network to predict the scene of a more reasonable scale, yielding the best performance.

Once the 3D Gaussians are reconstructed in the canonical space, we leverage it for both novel view synthesis (NVS) and pose estimation. For pose estimation, we introduce a two-stage pipeline: first, we obtain an initial pose estimate by applying the PnP algorithm (Hartley & Zisserman, 2003) to the

Gaussian centers. This rough estimate is then refined by rendering the scene at the estimated pose and optimizing the alignment with the input view using photometric loss.

Extensive experimental results demonstrate that our method performs impressively in both NVS and pose estimation tasks. For NVS, we show for the first time that, when trained on the same dataset under the same settings, a pose-free method can outperform pose-dependent methods, especially when the overlap between the two input images is small. In terms of pose estimation, our method significantly outperforms prior SOTA across multiple benchmarks. Additionally, NoPoSplat generalizes well to out-of-distribution data. Since our method does not require camera poses for input images, it can be applied to user-provided images to reconstruct the underlying 3D scene and render novel views. To illustrate this, we apply our model to sparse image pairs captured with mobile phones, as well as to sparse frames extracted from videos generated by Sora (OpenAI, 2024).

The main contributions of this work are:

- We propose NoPoSplat, a feed-forward network that reconstructs 3D scenes parameterized by 3D Gaussians from unposed sparse-view inputs, and demonstrate that it can be trained using photometric loss alone.
- We investigate the scale ambiguity issue of the reconstructed Gaussians, and solve this problem by introducing a camera intrinsic token embedding.
- We design a two-stage pipeline that estimates accurate relative camera poses using the reconstructed Gaussian field.

## 2 RELATED WORK

**Generalizable 3D Reconstruction and View Synthesis.** NeRF (Mildenhall et al., 2020) and 3D Gaussian Splatting (3DGS) (Kerbl et al., 2023) have significantly advanced 3D reconstruction and novel view synthesis. However, these methods typically require dense posed images (*e.g.*, hundreds) as input and minutes to hours of per-scene optimization, even with improved techniques (Chen et al., 2022; Fridovich-Keil et al., 2023; Müller et al., 2022). This limits their practical applications. To address these limitations, recent approaches focus on generalizable 3D reconstruction and novel view synthesis from sparse inputs (Yu et al., 2021; Wang et al., 2021a; Xu et al., 2024b; Johari et al., 2022; Charatan et al., 2024; Chen et al., 2024). They typically incorporate task-specific backbones that leverage geometric information to enhance scene reconstruction. For instance, MVSNeRF (Chen et al., 2021) and MuRF (Xu et al., 2024b) build cost volumes to aggregate multi-view information, while pixelSplat (Charatan et al., 2024) employs epipolar geometry for improved depth estimation. However, these geometric operations often require camera pose input and sufficient camera pose overlap among input views. In contrast, our network is based solely on a vision transformer (Dosovitskiy et al., 2021) without any geometric priors, making it pose-free and more effective in handling scenes with large camera baselines. Some recent approachs (Xu et al., 2024c; Tang et al., 2024; Zhang et al., 2025) also employ a simple backbone, but still require the camera pose as input.

**Pose-Free 3D Scene Reconstruction.** Classical NeRF or 3DGS-based methods require accurate camera poses of input images, typically obtained through Structure from Motion (SfM) methods like COLMAP (Schonberger & Frahm, 2016), which complicates the overall process. Recent per-scene optimization works (Wang et al., 2021b; Lin et al., 2021; Chng et al., 2022; Truong et al., 2023) jointly optimize camera poses and neural scene representations, but they still require rough pose initialization or are limited to small motions. Others (Bian et al., 2023; Fu et al., 2024) adopt incremental approaches from (Zhu et al., 2022; 2024; Matsuki et al., 2024), but they only allow image/video sequences as input. Moreover, for generalizable sparse-view methods, requiring camera poses during inference presents significant challenges, as these poses are often unavailable in real-world applications during testing. Although sparse-view pose estimation methods (Wang et al., 2024b; Zhang et al., 2022; Lin et al., 2023; Zhang et al., 2024; Jiang et al., 2024a) can be used, they are prone to failure in textureless regions or when images lack sufficient overlap. Additionally, the initial pose estimation stage introduces noise into the reconstruction process. Some recent generalizable pose-free novel view synthesis methods (Chen & Lee, 2023; Smith et al., 2023; Hong et al., 2024a; Xu et al., 2024a) attempt to address this but typically break the task into two stages: first estimate camera poses, then construct the scene representation. This two-stage process still lags behind pose-required methods because the initial pose estimation introduces noise, degrading reconstruction quality. Some methods (Sajjadi et al., 2022; Nagoor Kani et al., 2024) transform images into latent representations. However, these methods exhibit poor performance and cannot be used for

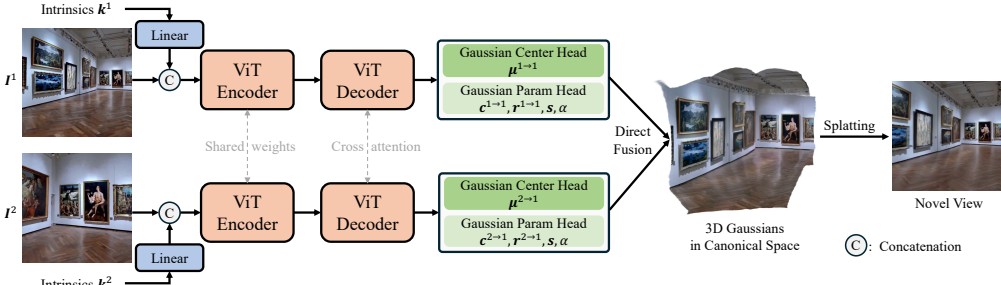

Figure 3: **Overview of NoPoSplat.** We directly predict Gaussians in a canonical space from a feed-forward network to represent the underlying 3D scene from the unposed sparse images. For simplicity, we use a two-view setup as an example, and the RGB shortcut is omitted from the figure.

pose estimation, limiting their applications. LEAP (Jiang et al., 2024b) and PF-LRM (Wang et al., 2024a) share a similar approach of lifting multi-view unposed images to a NeRF representation. However, their low rendering efficiency and resolution constraints hinder training effectiveness and limit scalability for complex scene-level reconstruction. In contrast, our method completely eliminates camera poses by directly predicting 3D Gaussians in a canonical space, avoiding potential noise in pose estimation and achieving better scene reconstruction.

One concurrent work, Splatt3R (Smart et al., 2024) , also predicts Gaussians in a global coordinate system but relies on the frozen MASt3R (Leroy et al., 2024) model for Gaussian centers. This is unsuitable for novel view synthesis, as MASt3R struggles to merge scene content from different views smoothly. Moreover, in contrast to us, Splatt3R requires ground truth depth during training, so it cannot utilize large-scale video data without depths or ground truth metric camera poses. Another concurrent work, ReconX (Liu et al., 2024), also reconstructs 3D Gaussians from unposed images. However, this approach relies on per-scene optimization and multi-stage inference, making it significantly less efficient compared to our feed-forward pipeline.

## 3 METHOD

### 3.1 PROBLEM FORMULATION

Our method takes as input sparse *unposed* multi-view images and corresponding camera intrinsic parameters $\{\boldsymbol{I}^v, \boldsymbol{k}^v\}_{v=1}^V$, where $V$ is the number of input views, and learn a feed-forward network $f_{\boldsymbol{\theta}}$ with learnable parameters $\theta$. The network maps the input unposed images to 3D Gaussians in a *canonical* 3D space, representing the underlying scene geometry and appearance. Formally, we aim to learn the following mapping:

$$f_{\boldsymbol{\theta}} : \{(\boldsymbol{I}^v, \boldsymbol{k}^v)\}_{v=1}^V \mapsto \left\{\cup \left(\boldsymbol{\mu}_j^v, \alpha_j^v, \boldsymbol{r}_j^v, \boldsymbol{s}_j^v, \boldsymbol{c}_j^v\right)\right\}_{j=1,\dots,H\times W}^{v=1,\dots,V}, \qquad (1)$$

where the right side represents the Gaussian parameters (Kerbl et al., 2023). Specifically, we have the center position $\boldsymbol{\mu} \in \mathbb{R}^3$, opacity $\alpha \in \mathbb{R}$, rotation factor in quaternion $\boldsymbol{r} \in \mathbb{R}^4$, scale $\boldsymbol{s} \in \mathbb{R}^3$, and spherical harmonics (SH) $\boldsymbol{c} \in \mathbb{R}^k$ with $k$ degrees of freedom. Note that, in line with common practice in pose-free scene reconstruction methods (Hong et al., 2024a; Fu et al., 2024; Smith et al., 2023; Chen & Lee, 2023), we assume having camera intrinsic parameters $\boldsymbol{k}$ as input, as they are generally available from modern devices (Arnold et al., 2022).

By training on large-scale datasets, our method can generalize to novel scenes without any optimization. The resulting 3D Gaussians in the canonical space directly enable two tasks: a) novel view synthesis given the target camera transformation relative to the first input view, and b) relative pose estimation among different input views. Next, we will introduce our overall pipeline.

### 3.2 PIPELINE

Our method, illustrated in Fig. 3, comprises three main components: an encoder, a decoder, and Gaussian parameter prediction heads. Both encoder and decoder utilize pure Vision Transformer (ViT) structures, without injecting any geometric priors (e.g. epipolar constraints employed in pixelSplat (Charatan et al., 2024), or cost volumes in MVSplat (Chen et al., 2024)). Interestingly, we demonstrate in Sec. 4 that such a simple ViT network shows competitive or superior performance over those dedicated backbones incorporating these geometric priors, especially in scenarios with

limited content overlap between input views. This advantage stems from the fact that such geometric priors typically necessitate substantial overlap between input cameras to be effective.

**ViT Encoder and Decoder.** The RGB images are patchified and flattened into sequences of image tokens, and then concatenated with an intrinsic token (details in Sec. 3.4). The concatenated tokens from each view are then fed into a ViT (Dosovitskiy et al., 2021) encoder separately. The encoder shares the same weights for different views. Next, the output features from the encoder are fed into a ViT decoder module, where features from each view interact with those from all other views through cross-attention layers in each attention block, facilitating multi-view information integration.

**Gaussian Parameter Prediction Heads.** To predict the Gaussian parameters, we employ two prediction heads based on the DPT architecture (Ranftl et al., 2021). The first head focuses on predicting the Gaussian center positions and utilizes features extracted exclusively from the transformer decoder. The second head predicts the remaining Gaussian parameters and, in addition to the ViT decoder features, also takes the RGB image as input. Such RGB image shortcut ensures the direct flow of texture information, which is crucial for capturing fine texture details in 3D reconstruction. This approach compensates for the high-level features output by the ViT decoder, downsampled by a factor of 16, which are predominantly semantic and lack detailed structural information. With these prediction heads in place, we now analyze how our method differs from previous approaches in terms of the output Gaussian space and the advantages this brings.

## 3.3 ANALYSIS OF THE OUTPUT GAUSSIAN SPACE

While our method shares a similar spirit with previous works (Charatan et al., 2024; Zheng et al., 2024; Szymanowicz et al., 2024) in predicting pixelwise Gaussians for input images, we differ in the output Gaussian space. In this section, we first discuss the local-to-global Gaussian space in prior methods and its inherent limitations, and introduce our canonical Gaussian space.

**Baseline: Local-to-Global Gaussian Space.** Previous methods first predict the corresponding depth of each pixel, then lift the predicted Gaussian parameters to a Gaussian primitive in the local coordinate system of each individual frame using the predicted depth and the camera intrinsics. These local Gaussians are then transformed into a world coordinate system using the given camera poses $[\boldsymbol{R}^v \mid \boldsymbol{t}^v]$ for each input view. Finally, all transformed Gaussians are fused to represent the underlying scene.

However, this strategy has two main issues: a) Transforming Gaussians from local to world coordinates requires accurate camera poses, which are difficult to obtain in real-world scenarios with sparse input views. b) The transform-then-fuse method struggles to combine 3D Gaussians from different views into a cohesive global representation, especially when the overlap among input views is small, or when generalizing to out-of-distribution data (as shown in Fig. 5 and Fig. 6).

**Proposed: Canonical Gaussian Space.** In contrast, we directly output Gaussians of different views in a canonical coordinate system. Specifically, we anchor the first input view as the global reference coordinate system. Therefore, the camera pose for the first input view is $[\boldsymbol{U} \mid \boldsymbol{0}]$, where $\boldsymbol{U}$ is the unit/identity matrix for rotation, and $\boldsymbol{0}$ is the zero translation vector. The network outputs Gaussians under this canonical space for all input views. Formally, for each input view, we predict the set $\left\{\boldsymbol{\mu}_j^{v\rightarrow1}, \boldsymbol{r}_j^{v\rightarrow1}, \boldsymbol{c}_j^{v\rightarrow1}, \alpha_j, \boldsymbol{s}_j\right\}$, where the superscript $v \rightarrow 1$ denotes that the Gaussian parameters corresponding to pixel $\boldsymbol{p}_j$ in view $v$, are under the local camera coordinate system of view 1.

Predicting directly under the canonical space offers several benefits. First, the network learns to fuse different views directly within the canonical space, eliminating the need for camera poses. Second, bypassing the transform-then-fuse step results in a cohesive global representation, which further unlocks the application of pose estimation for input unposed views.

## 3.4 CAMERA INTRINSICS EMBEDDING

As discussed in Eq. 1, our network inputs also include the camera intrinsics $\boldsymbol{k}$ of each input view. This is required to resolve the inherent scale misalignment and provide essential geometric information that improves 3D reconstruction quality (*cf*. "No Intrinsics" in Fig. 8 and Tab. 5). Although intrinsic embedding has been employed in previous NeRF-based methods, both pose-free (Wang et al., 2024a) and pose-required (Hong et al., 2024b), its necessity for generalizable pose-free Gaussian Splatting prediction and optimal embedding approaches remain unexplored. Therefore, we introduce three different encoding strategies for injecting camera intrinsics into our model.

**Global Intrinsic Embedding - Addition.** A straightforward strategy is to feed camera intrinsics $\boldsymbol{k} = [f_x, f_y, c_x, c_y]$ into a linear layer to obtain a global feature. This feature is broadcast and added to the RGB image features after the patch embedding layer of the ViT.

**Global Intrinsic Embedding - Concat.** After obtaining the global feature, we instead treat it as an additional intrinsic token, and concatenate it with all image tokens.

**Dense Intrinsic Embedding.** For each pixel $\boldsymbol{p}_j$ in the input view, we can obtain the camera ray direction as $\boldsymbol{K}^{-1}\boldsymbol{p}_j$, where $\boldsymbol{K}$ is the matrix form of $\boldsymbol{k}$. These per-pixel camera rays are then converted using spherical harmonics to higher-dimension features and concatenated with the RGB image as the network input. Note that the pixel-wise embedding can be viewed as a simplification of the widely-used Plücker ray embedding (Xu et al., 2023; Tang et al., 2024), as it does not require camera extrinsic information.

By default, we adopt the "global intrinsic embedding - concat" option since it is not only an elegant way to inject camera intrinsic into the network but also yields the best performance (see Tab. 5).

## 3.5 TRAINING AND INFERENCE

**Training.** The predicted 3D Gaussians are rendered at novel viewpoints using their corresponding ground-truth camera poses. Our network is end-to-end trained using ground truth target RGB images as supervision. Following Chen et al. (2024), we also use a linear combination of MSE and LPIPS (Zhang et al., 2018) loss with weights of 1 and 0.05, respectively.

**Relative Pose Estimation.** As mentioned in Sec. 3.3, since our 3D Gaussians are in the canonical space, they can be directly used for relative pose estimation. To facilitate efficient pose estimation, we propose a two-step approach. First, we estimate the initial related camera pose of the input two views using the PnP algorithm (Hartley & Zisserman, 2003) with RANSAC (Fischler & Bolles, 1981), given the Gaussian centers of the output Gaussians in world coordinates. This step is highly efficient and done in milliseconds. Next, while keeping Gaussian parameters frozen, we refine the initial pose from the first step by optimizing the same photometric losses used for model training, along with the structural part of the SSIM loss (Wang et al., 2004). During the optimization, we calculate the camera Jacobian to reduce the computational overhead associated with automatic differentiation and decrease optimization time as in Matsuki et al. (2024).

**Evaluation-Time Pose Alignment.** Given unposed image pairs, our method learns to reconstruct a plausible 3D scene that aligns with the given inputs. However, 3D scene reconstruction with just two input views is inherently ambiguous as many different scenes can produce the same two images. As a result, though the scene generated by our method successfully explains the input views, it might not be exactly the same as the ground truth scene in the validation dataset. Thus, to fairly compare with other baselines, especially ones that utilize ground truth camera poses(Du et al., 2023; Charatan et al., 2024), we follow previous pose-free works (Wang et al., 2021b; Fan et al., 2024) and optimize the camera pose for the target view. Specifically, for each evaluation sample, we first reconstruct 3D Gaussians using the proposed method. We then freeze the Gaussians and optimize the target camera pose such that the rendered image from the target view closely matches the ground truth image. It is important to note that this optimization is solely for evaluation purposes and is not required when applying our method in real-world scenarios (*e.g.*, Fig. 7).

## 4 EXPERIMENTS

**Datasets.** To evaluate novel view synthesis, we follow the setting in (Charatan et al., 2024; Chen et al., 2024) and train and evaluate our method on RealEstate10k (RE10K) (Zhou et al., 2018) and ACID (Liu et al., 2021) datasets separately. RE10K primarily contains indoor real estate videos, while ACID features nature scenes captured by aerial drones. Both include camera poses calculated using COLMAP (Schonberger & Frahm, 2016). We adhere to the official train-test split as in previous works (Charatan et al., 2024; Hong et al., 2024a). To further scale up our model (denoted as **Ours***), we also combine RE10K with DL3DV (Ling et al., 2024), which is an outdoor dataset containing 10K videos, which includes a wider variety of camera motion patterns than RE10K.

To assess the method's capability in handling input images with varying camera overlaps, we generate input pairs for evaluation that are categorized based on the ratio of image overlap: small (5% - 30%), medium (30% - 55%), and large (55% - 80%), determined using SOTA dense feature matching method, RoMA (Edstedt et al., 2024). More details are provided in the appendix.

Table 1: **Novel view synthesis performance comparison on the RealEstate10k (Zhou et al., 2018) dataset**. Our method largely outperforms previous pose-free methods on all overlap settings, and even outperforms SOTA pose-required methods, especially when the overlap is small.

| | Method | Small | | | Medium | | | Large | | | Average | | |
|---|---|---|---|---|---|---|---|---|---|---|---|---|---|
| | | PSNR↑ | SSIM↑ | LPIPS↓ | PSNR↑ | SSIM↑ | LPIPS↓ | PSNR↑ | SSIM↑ | LPIPS↓ | PSNR↑ | SSIM↑ | LPIPS↓ |
| *Pose-Required* | pixelNeRF | 18.417 | 0.601 | 0.526 | 19.930 | 0.632 | 0.480 | 20.869 | 0.639 | 0.458 | 19.824 | 0.626 | 0.485 |
| | AttnRend | 19.151 | 0.663 | 0.368 | 22.532 | 0.763 | 0.269 | 25.897 | 0.845 | 0.186 | 22.664 | 0.762 | 0.269 |
| | pixelSplat | 20.263 | 0.717 | 0.266 | 23.711 | 0.809 | 0.181 | 27.151 | 0.879 | 0.122 | 23.848 | 0.806 | 0.185 |
| | MVSplat | 20.353 | 0.724 | 0.250 | 23.778 | 0.812 | 0.173 | 27.408 | **0.884** | **0.116** | 23.977 | 0.811 | 0.176 |
| *Pose-Free* | Splatt3R | 14.352 | 0.475 | 0.472 | 15.529 | 0.502 | 0.425 | 15.817 | 0.483 | 0.421 | 15.318 | 0.490 | 0.436 |
| | CoPoNeRF | 17.393 | 0.585 | 0.462 | 18.813 | 0.616 | 0.392 | 20.464 | 0.652 | 0.318 | 18.938 | 0.619 | 0.388 |
| | **Ours** | **22.514** | **0.784** | **0.210** | **24.899** | **0.839** | **0.160** | **27.411** | 0.883 | 0.119 | **25.033** | **0.838** | **0.160** |

Table 2: **Novel view synthesis performance comparison on the ACID (Liu et al., 2021) dataset**.

| | Method | Small | | | Medium | | | Large | | | Average | | |
|---|---|---|---|---|---|---|---|---|---|---|---|---|---|
| | | PSNR↑ | SSIM↑ | LPIPS↓ | PSNR↑ | SSIM↑ | LPIPS↓ | PSNR↑ | SSIM↑ | LPIPS↓ | PSNR↑ | SSIM↑ | LPIPS↓ |
| *Pose-Required* | pixelNeRF | 19.376 | 0.535 | 0.564 | 20.339 | 0.561 | 0.537 | 20.826 | 0.576 | 0.509 | 20.323 | 0.561 | 0.533 |
| | AttnRend | 20.942 | 0.616 | 0.398 | 24.004 | 0.720 | 0.301 | 27.117 | 0.808 | 0.207 | 24.475 | 0.730 | 0.287 |
| | pixelSplat | 22.053 | 0.654 | 0.285 | 25.460 | 0.776 | 0.198 | **28.426** | **0.853** | 0.140 | 25.819 | 0.779 | 0.195 |
| | MVSplat | 21.392 | 0.639 | 0.290 | 25.103 | 0.770 | 0.199 | 28.388 | 0.852 | **0.139** | 25.512 | 0.773 | 0.196 |
| *Pose-Free* | CoPoNeRF | 18.651 | 0.551 | 0.485 | 20.654 | 0.595 | 0.418 | 22.654 | 0.652 | 0.343 | 20.950 | 0.606 | 0.406 |
| | **Ours** | **23.087** | **0.685** | **0.258** | **25.624** | **0.777** | **0.193** | 28.043 | 0.841 | 0.144 | **25.961** | **0.781** | **0.189** |

Furthermore, for zero-shot generalization, we also test on DTU (Jensen et al., 2014) (object-centric scenes), ScanNet (Dai et al., 2017) and ScanNet++ (Yeshwanth et al., 2023) (indoor scenes with different camera motion and scene types from the RE10K). We also demonstrate our approach on in-the-wild mobile phone capture, and images generated by a text-to-video model (OpenAI, 2024).

**Evaluation Metrics.** We evaluate novel view synthesis with the commonly used metrics: PSNR, SSIM, and LPIPS (Zhang et al., 2018). For pose estimation, we report the area under the cumulative pose error curve (AUC) with thresholds of $5°$, $10°$, $20°$ (Sarlin et al., 2020; Edstedt et al., 2024).

**Baselines.** We compare against SOTA sparse-view generalizable methods on novel view synthesis: 1) *Pose-required*: pixelNeRF (Yu et al., 2021), AttnRend (Du et al., 2023), pixelSplat (Charatan et al., 2024), and MVSplat (Chen et al., 2024); 2) *Pose-free*: DUSt3R (Wang et al., 2024b), MASt3R (Leroy et al., 2024), Splatt3R (Smart et al., 2024), CoPoNeRF (Hong et al., 2024a), and RoMa (Edstedt et al., 2024). For relative pose estimation, we also compare against methods in 2).

**Implementation details.** We use PyTorch, and the encoder is a vanilla ViT-large model with a patch size of 16, and the decoder is ViT-base. We initialize the encoder/decoder and Gaussian center head with the weights from MASt3R, while the remaining layers are initialized randomly. Note that, as shown in Tab. 7, our method can also be trained with only RGB supervision–without pre-trained weight from MASt3R–and still achieve similar performance. We train models at two different resolutions, $256 \times 256$ and $512 \times 512$. For a fair comparison with baseline models, we report all quantitative results and baseline comparisons under $256 \times 256$. However, qualitative results for the $512 \times 512$ model are presented in the supplementary video and Fig. 7. Additional details on model weight initialization and training resolution can be found in the appendix.

## 4.1 EXPERIMENTAL RESULTS AND ANALYSIS

**Novel View Synthesis.** As demonstrated in Tab. 1, Tab. 2, and Fig. 4, NoPoSplat significantly outperforms all SOTA pose-free approaches. Note that DUSt3R (and MASt3R) struggle to fuse input views coherently due to their reliance on per-pixel depth loss, a limitation Splatt3R also inherits from its frozen MASt3R module. On the other hand, we achieve competitive performance over SOTA pose-required methods (Charatan et al., 2024; Chen et al., 2024), and even outperform them when the overlap between input images is small, as shown in Fig.4 first row (the left side of Fig. 4 displays the overlap ratio). This clearly shows the advantages of our model's 3D Gaussians prediction in the canonical space over baseline methods' transform-then-fuse strategy, as discussed in Sec. 3.3.

**Relative Pose Estimation.** The proposed method can be applied to pose estimation between input views on three diverse datasets. Our method is trained either on RE10K (denoted as Ours) or a combination of RE10K and DL3DV (denoted as Ours*). Tab. 3 shows that the performance consistently improves when scaling up training with DL3DV involved. This can be attributed to the greater variety of camera motions in DL3DV over in RE10K. It is worth noting that our method

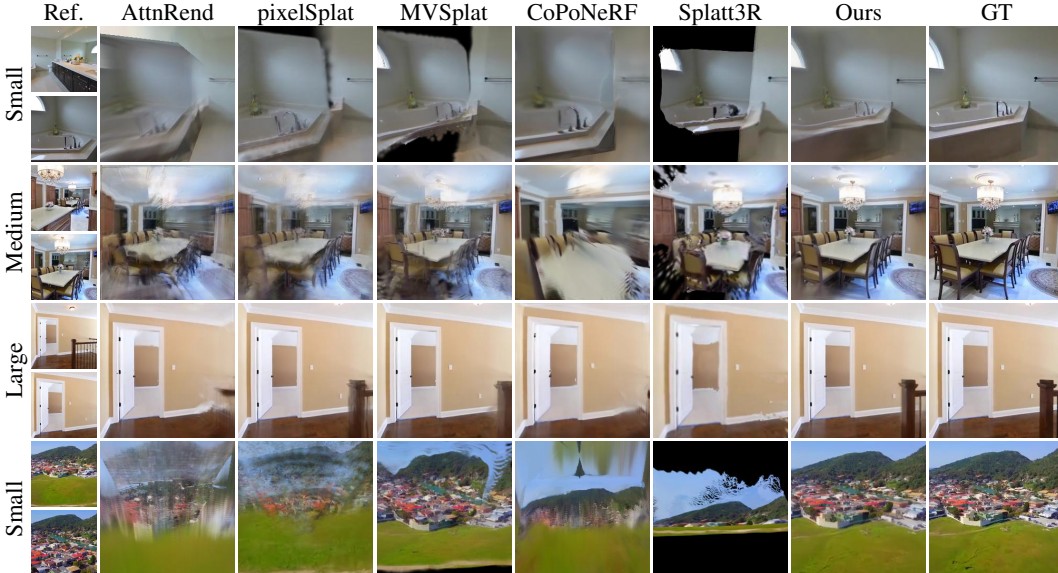

Figure 4: **Qualitative comparison on RE10K (top three rows) and ACID (bottom row).** Compared to baselines, we obtain: 1) more coherent fusion from input views, 2) superior reconstruction from limited image overlap, 3) enhanced geometry reconstruction in non-overlapping regions.

Table 3: **Pose estimation performance in AUC with various thresholds on RE10k, ACID, and ScanNet-1500 (Dai et al., 2017; Sarlin et al., 2020).** Our method achieves the best results across all datasets. Notably, our model is not trained on ACID or ScanNet. Furthermore, our method does not require an explicit matching loss during training, meaning no ground truth depth is necessary.

| Method | RE10k | | | ACID | | | ScanNet-1500 | | |
|---|---|---|---|---|---|---|---|---|---|
| | 5° ↑ | 10° ↑ | 20° ↑ | 5° ↑ | 10° ↑ | 20° ↑ | 5° ↑ | 10° ↑ | 20° ↑ |
| CoPoNeRF | 0.161 | 0.362 | 0.575 | 0.078 | 0.216 | 0.398 | - | - | - |
| DUSt3R | 0.301 | 0.495 | 0.657 | 0.166 | 0.304 | 0.437 | 0.221 | 0.437 | 0.636 |
| MASt3R | 0.372 | 0.561 | 0.709 | 0.234 | 0.396 | 0.541 | 0.159 | 0.359 | 0.573 |
| RoMa | 0.546 | 0.698 | 0.797 | 0.463 | 0.588 | 0.689 | 0.270 | 0.492 | 0.673 |
| **Ours** (RE10k) | 0.672 | 0.792 | 0.869 | 0.454 | 0.591 | 0.709 | 0.264 | 0.473 | 0.655 |
| **Ours*** (RE10k+DL3DV) | **0.691** | **0.806** | **0.877** | **0.486** | **0.617** | **0.728** | **0.318** | **0.538** | **0.717** |

Table 4: **Out-of-distribution performance comparison.** Our method shows superior performance when zero-shot evaluation on DTU and ScanNet++ using the model solely trained on RE10k.

| Training Data | Method | DTU | | | ScanNet++ | | |
|---|---|---|---|---|---|---|---|
| | | PSNR ↑ | SSIM ↑ | LPIPS ↓ | PSNR ↑ | SSIM ↑ | LPIPS ↓ |
| ScanNet++ | Splatt3R | 11.628 | 0.325 | 0.499 | 13.270 | 0.507 | 0.445 |
| RealEstate10K | pixelSplat | 11.551 | 0.321 | 0.633 | 18.434 | 0.719 | 0.277 |
| | MVSplat | 13.929 | 0.474 | 0.385 | 17.125 | 0.686 | 0.297 |
| | Ours | **17.899** | **0.629** | **0.279** | **22.136** | **0.798** | **0.232** |

even shows superior zero-shot performance on ACID and ScanNet-1500, even better than the SOTA method RoMa that is trained on ScanNet. This indicates not only the efficacy of our pose estimation approach, but also the quality of our output 3D geometry. The next part verifies this point.

**Geometry Reconstruction.** Our method also outputs noticeably better 3D Gaussians and depths over SOTA pose-required methods, as shown in Fig. 5. Looking closely, MVSplat not only suffers from the misalignment in the intersection regions of two input images (indicated by blue arrows), but also distortions or incorrect geometry in regions without sufficient overlap (indicated by magenta arrows). These issues are largely due to the noises introduced in their transform-then-fuse pipeline. Our method directly predicts Gaussians in the canonical space, which faithfully solves this problem.

**Cross-Dataset Generalization.** We also evaluate the zero-shot performance of the model, where we train exclusively on RealEstate10k and directly apply it to ScanNet++ (Yeshwanth et al., 2023) and DTU (Jensen et al., 2014) datasets. The results in Fig. 6 and Tab. 4 indicate that NoPoSplat demon-

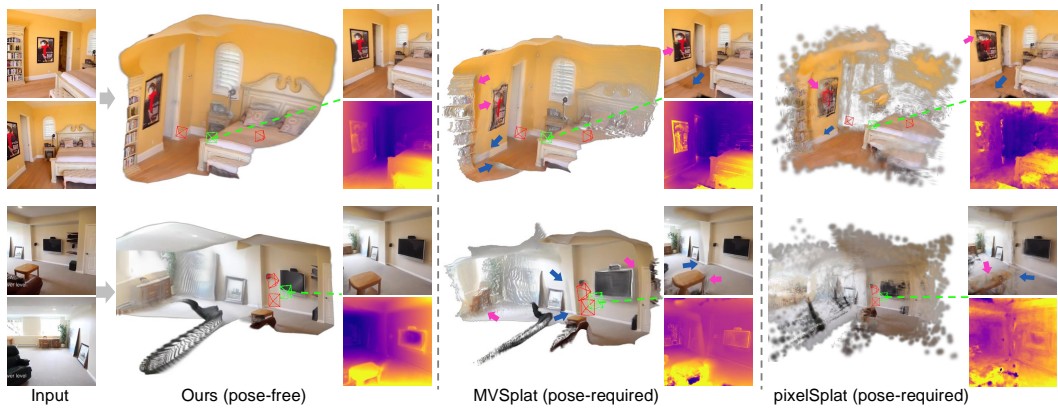

Figure 5: **Comparisons of 3D Gaussian and rendered results.** The red and green indicate input and target camera views, and the rendered image and depths are shown on the right side. The magenta and blue arrows correspond to the distorted or misalignment regions in baseline 3DGS. The results show that even without camera poses as input, our method produces higher-quality 3D Gaussians resulting in better color/depth rendering over baselines.

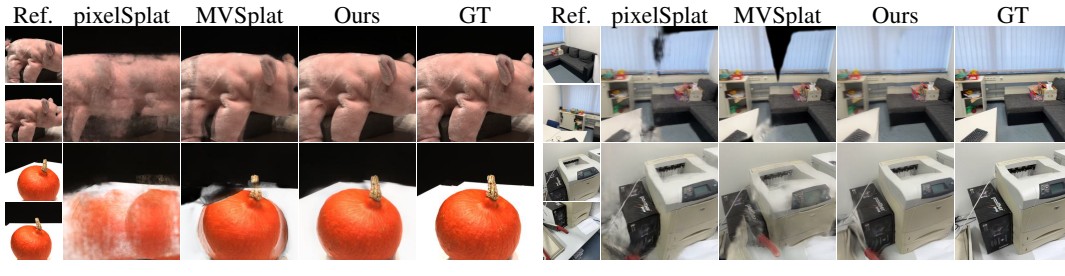

(a) Cross-Dataset Generalize: RE10K → DTU     (b) Cross-Dataset Generalize: RE10K → ScanNet++

Figure 6: **Cross-dataset generalization.** Our model can better zero-shot transfer to out-of-distribution data than SOTA pose-required methods.

strates superior performance on out-of-distribution data compared to SOTA pose-required methods. This advantage arises primarily from our minimal geometric priors in the network structure, allowing it to adapt effectively to various types of scenes. Notably, our method outperforms Splatt3R even on ScanNet++, where Splatt3R was trained.

**Model Efficiency.** As shown on the right, our method can predict 3D Gaussians from two $256 \times 256$ input images in 0.015 seconds (66 fps), which is around $5\times$ and $2\times$ faster than pixelSplat and MVSplat, on the same RTX 4090 GPU. This further shows the benefits of using a standard ViT without incorporating additional geometric operations.

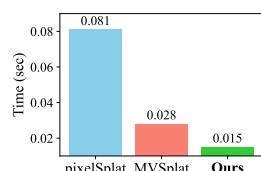

**Apply to In-the-Wild Unposed Images.** One of the most important advantages of our method is that it can directly generalize to in-the-wild unposed images. we test on two types of data: images casually taken with mobile phones, and image frames extracted from videos generated by Sora (OpenAI, 2024). Results in Fig. 7 show that our method can be potentially applied for text/image to 3D scene generations, *e.g.*, one can first generate sparse scene-level multi-view images using text/image to multi-image/video models (Gao et al., 2024; OpenAI, 2024), then feed those extracted unposed images to our model and obtain 3D models.

## 4.2 ABLATION STUDIES

**Ablation on Output Gaussian Space.** To demonstrate the effectiveness of our canonical Gaussian prediction, we compare it with the transform-then-fuse pipeline commonly used by pose-required methods (Charatan et al., 2024; Chen et al., 2024). Specifically, it first predicts Gaussians in each local camera coordinate system which was transformed to the world coordinate using camera poses. For fair comparisons, both methods use the same backbone and head but differ in the prediction of Gaussian space. The results in row (f) of Tab. 5 show that our pose-free canonical space predic-

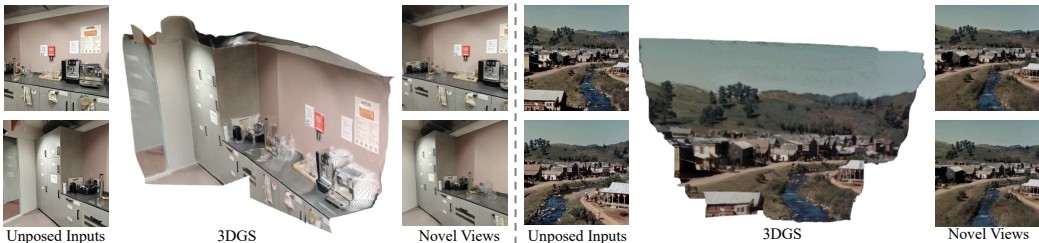

| Unposed Inputs | 3DGS | Novel Views | Unposed Inputs | 3DGS | Novel Views |

(a) Input Images taken with mobile phones  (b) Input Images generated by SORA

Figure 7: **In-the-wild Data.** We present the results of applying our method to in-the-wild data, including real-world photos taken with mobile phones and multi-view images extracted from videos generated by the Sora text-to-video model. See the video supplementary for more results.

tion method outperforms such pose-required strategy. Fig. 8 illustrates that the transform-then-fuse strategy leads to the ghosting artifacts in the rendering, because it struggles to align two separate Gaussians of two input views when transformed to a global coordinate.

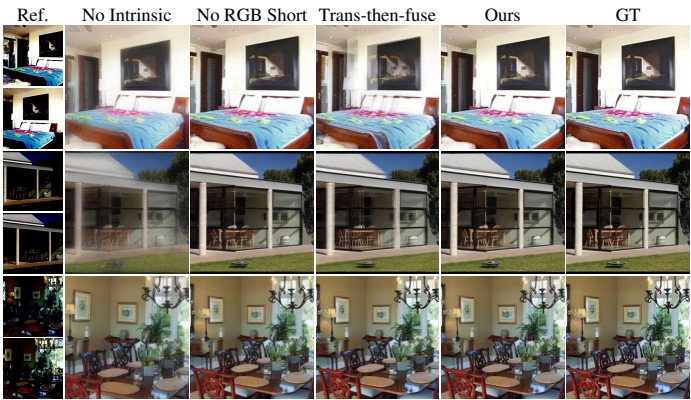

Figure 8: **Ablations.** No intrinsic results in blurriness due to scale misalignment. Without the RGB image shortcut, the rendered images are blurry in the texture-rich areas. Using the transform-then-fuse strategy causes ghosting problem.

| Num | Design | PSNR | SSIM | LPIPS |
|-----|--------|------|------|-------|
| (a) | Ours | **25.033** | **0.838** | **0.160** |
| (b) | No intrinsic emb. | 23.543 | 0.780 | 0.186 |
| (c) | w/ dense emb. | 24.809 | 0.832 | 0.166 |
| (d) | w/ global emb. - add. | 24.952 | 0.836 | 0.161 |
| (e) | No RGB shortcut | 24.279 | 0.810 | 0.183 |
| (f) | Transform-then-fuse | 24.632 | 0.834 | 0.167 |
| (g) | 3 input views | 26.619 | 0.872 | 0.127 |

Table 5: **Ablations.** intrinsic embeddings are vital for performance and using intrinsic tokens performs the best. Adding the RGB image shortcut also improves the quality of rendered images. Our method achieves better performance compared with the pose-required per-local-view Gaussian field prediction method.

**Ablation on Camera Intrinsic Embedding.** Here we study three intrinsic encodings described in Sec. 3.4 as well as inputting no intrinsic information. First, we can see in Tab. 5 row (b) and Fig. 8 that no intrinsic encodings lead to blurry results as the scale ambiguity confuses the learning process of the model. Second, we notice that the intrinsic token constantly performs the best marginally among these three proposed intrinsic encodings, so, we use it as our default intrinsic encoding.

**Importance of RGB Shortcut.** As discussed in Sec. 3.2, in addition to the low-res ViT features, we also input RGB images into the Gaussian parameter prediction head. As shown in Fig. 8, when there is no RGB shortcut, the rendered images are blurry in the texture-rich areas, see the quilt in row 1 and chair in row 3.

**Extend to More Input Views.** For fair comparisons with baselines, we primarily conduct experiments using two input view settings. Here we present results using three input views by adding an additional view between the two original input views. As shown in row (g) of Tab. 5 and Fig. 13, the performance significantly improves with the inclusion of the additional view. Results with more input views can be found in Fig. 10.

## 5 CONCLUSION

This paper introduces a simple yet effective pose-free method for generalizable sparse-view 3D reconstruction. By predicting 3D Gaussians directly in a canonical space from any given unposed multi-view images, we demonstrate superior performance in novel view synthesis and relative pose estimation. While our method currently applies only to static scenes, extending our pipeline to dynamic scenarios presents an interesting direction for future work.

ACKNOWLEDGMENTS

This work was supported as part of the Swiss AI Initiative by a grant from the Swiss National Supercomputing Centre (CSCS) under project ID a03 on Alps. Botao Ye was partially supported by the ETH AI Center. Ming-Hsuan Yang was supported by the Intelligence Advanced Research Projects Activity (IARPA) via Department of Interior/ Interior Business Center (DOI/IBC) contract number 140D0423C0074. We also thank David Charatan for helping us evaluate the performance of some of the previous methods.

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

## A   MORE IMPLEMENTATION DETAILS

**More Training Details.** We first describe the process for training the $256 \times 256$ model, which serves as the basis for all baseline comparisons. Following Chen et al. (2024), when training on RealEstate10K (Zhou et al., 2018) and ACID (Liu et al., 2021) separately, the model is trained on $2.4 \times 10^6$ input image pairs, randomly sampled from training video clips to ensure diverse input coverage. We employ the AdamW optimizer (Loshchilov & Hutter, 2018), setting the initial learning rate for the backbone to $2 \times 10^{-5}$ and other parameters to $2 \times 10^{-4}$. When trained on the combined RealEstate10k and DL3DV (Ling et al., 2024) datasets, we sample image pairs evenly from both datasets and double the training steps.

We then train the $512 \times 512$ model using the pre-trained weights of the $256 \times 256$ model. The $512 \times 512$ model is also trained on the combined RealEstate10K and DL3DV datasets, following the same training procedure. Since no fair baseline model is available for comparison, we focus primarily on the qualitative results of the $512 \times 512$ model. For instance, the results presented in Fig. 7 are obtained using the $512 \times 512$ model.

For the $256 \times 256$ version of the model, training was conducted on 8 NVIDIA GH200 GPUs (each with >80 GB memory) for approximately 6 hours. We also experimented with training our model on a single A6000 GPU (48 GB memory). While this setup required more time (approximately 90 hours), it achieved comparable performance (PSNR on RE10K: 25.018 with A6000 vs. 25.033 with GH200). For the $512 \times 512$ version, training was performed on 16 NVIDIA GH200 GPUs and required approximately one day.

**Evaluation Set Generation.** For the evaluation of the novel view synthesis task, we randomly select one image pair from each video sequence in the official test splits of RealEstate10K (Zhou et al., 2018) and ACID (Liu et al., 2021). We then randomly sample three frames between the input views as target views. For the pose estimation task, we use the same input image pair as in the novel view synthesis task, and employ the COLMAP pose as the ground truth.

To determine the overlap ratio of image pairs, we employ RoMa (Edstedt et al., 2024), a state-of-the-art dense feature matching method. The process for calculating the overlap ratio for an image pair $\left\{ \boldsymbol{I}^1, \boldsymbol{I}^2 \right\}$ is as follows:

1. Obtain dense image matching from $\boldsymbol{I}^1$ to $\boldsymbol{I}^2$ and vice versa.
2. Consider matching scores above 0.005 as valid.
3. Calculate the overlap ratio $r_{i \rightarrow j}$ for image $\boldsymbol{I}^i$ to image $\boldsymbol{I}^j$ as:
$$r_{i \rightarrow j} = \frac{\text{Number of valid matched pixels}}{\text{Total number of pixels}}$$
4. Compute $r_{1 \rightarrow 2}$ and $r_{2 \rightarrow 1}$.
5. Define the final overlap ratio as:
$$r_{\text{overlap}} = \min(r_{1 \rightarrow 2}, r_{2 \rightarrow 1})$$

The resulting evaluation set and the code used for its generation will be made publicly available to facilitate further research in this area.

**Statistics on the Evaluation Set.** The number of scenes in RealEstate10K (Zhou et al., 2018) for each overlap category is as follows: 1403 for small, 2568 for medium, and 1630 for large overlaps. In the ACID dataset (Liu et al., 2021), the number of scenes for each overlap category is: 249 for small, 644 for medium, and 448 for large overlaps.

**Baseline Setup for Pose Estimation Tasks.** For a fair comparison with RoMa (Edstedt et al., 2024), we first resize and center-crop the input images to match the dimensions used in our model ($256 \times 256$), then resize them to a coarse resolution ($560 \times 560$) and an upsampled resolution ($864 \times 864$) to fit RoMa's requirements. However, since DUSt3R (Wang et al., 2024b) and MASt3R (Leroy et al., 2024) are not trained with square image inputs, simply setting the input resolution to $256 \times 256$ leads to poor pose estimation performance. Therefore, we resize and center-crop their input images to $512 \times 256$, as officially supported by their models. Consequently, the image content visible to DUSt3R and MASt3R is greater than that seen by RoMa and our model.

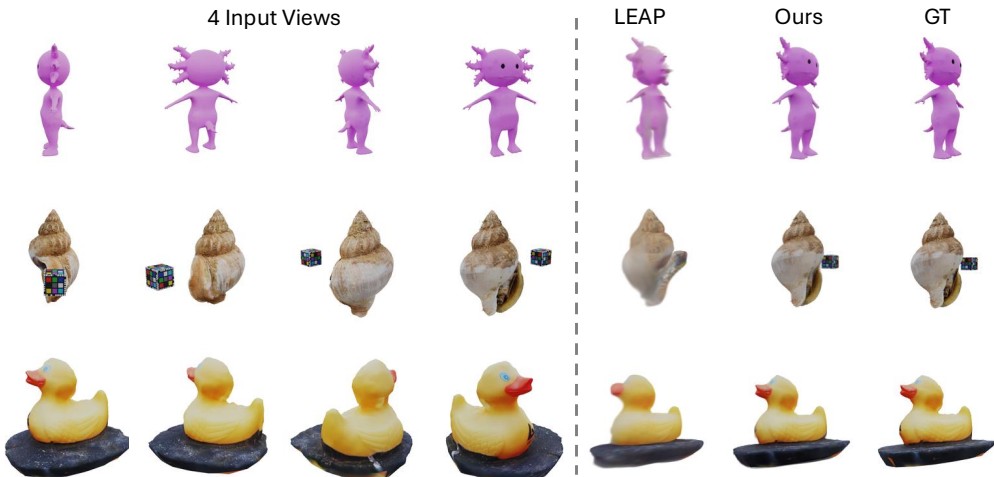

Figure 9: **Object-level comparison on Objaverse dataset.** Compared with the pose-free baseline method, LEAP, our method shows significantly better novel view synthesis results on object-level data.

Table 6: **Performance comparison on Objaverse dataset.**

|  | PSNR↑ | SSIM↑ | LPIPS↓ |
|---|---|---|---|
| LEAP (Jiang et al., 2024b) | 20.559 | 0.853 | 0.144 |
| **Ours** | **28.378** | **0.935** | **0.053** |

**Details on Pose Estimation.** After obtaining the coarse camera pose, we apply photometric loss-based optimization to refine it. This optimization is performed for 200 steps per input image pair with a learning rate of $5 \times 10^{-3}$, taking approximately 2 seconds. Notably, our pose estimation can also operate without the optimization refinement process, resulting in an acceptable performance degradation but decreasing inference time. As shown in Tab. 9, despite the performance drop, our method still performs comparably to RoMa, which is trained on ScanNet, even in a zero-shot setting.

**Details on Applying to In-the-Wild Data.** Our method requires camera intrinsic information as input for rendering novel views. When testing our model on out-of-distribution datasets, such as Tanks and Temples (Knapitsch et al., 2017), we simply use the ground truth intrinsic data provided by the dataset. When tested on photos taken with mobile phones, we extract the focal length from their EXIF metadata. For data generated by SORA (OpenAI, 2024), we set the focal length to $(H + W)/2$. We found that this heuristic setting works quite well, as our model is relatively robust to focal length variations after training.

## B  APPLY TO OBJECT-LEVEL DATA

To demonstrate our method's generalizability to object-level data, we evaluate its performance on the commonly used Objaverse dataset. Our experiments utilize four $256 \times 256$ unposed images as input for each object. Direct performance comparison with PF-LRM is not feasible due to its unavailable source code and evaluation dataset. Therefore, we compare our method with LEAP, the state-of-the-art open-source pose-free method for object-level reconstruction.

To ensure a fair comparison, we retrain LEAP (Jiang et al., 2024b) on the Objaverse dataset using their official implementation, maintaining identical training iterations and input image resolutions. The qualitative and quantitative results are presented in Fig. 9 and Tab. 6 respectively. Our approach demonstrates significant performance improvements over LEAP, with the PSNR increasing from 20.559 to 28.378, highlighting robust generalization capabilities for object-level reconstruction.

Table 7: **Ablation on different weight initialization**. The results show that our method effectively learns pose-free inference capabilities during training, with appropriate weight initialization further enhancing performance. Notably, even with random initialization, our method significantly outperforms the pose-free baseline (CoPoNeRF). Moreover, utilizing CroCov2 and DINOv2 pre-trained weights enables our method to surpass the previous SOTA pose-required method (MVSplat), despite neither initialization method providing prior knowledge of pose or depth information.

| | PSNR↑ | SSIM↑ | LPIPS↓ |
|---|---|---|---|
| MASt3R | **25.033** | **0.838** | **0.160** |
| DUSt3R | 24.553 | 0.823 | 0.169 |
| CroCov2 | 24.559 | 0.818 | 0.171 |
| DINOv2 | 24.094 | 0.812 | 0.176 |
| Random | 23.487 | 0.779 | 0.189 |

Table 8: **Ablation on adding additional pose condition.** We incorporate Plucker ray pose embeddings into our pose-free model as additional input. The results show a small performance gap after adding the embeddings, indicating the strong capability of our method in estimating pose.

| Init Weights | Pose Condition | PSNR↑ | SSIM↑ | LPIPS↓ |
|---|---|---|---|---|
| MASt3R | Yes | 25.033 | 0.838 | 0.160 |
| | No | 25.080 | 0.844 | 0.158 |
| Random | Yes | 23.708 | 0.788 | 0.173 |
| | No | 23.487 | 0.779 | 0.189 |

## C    MORE EXPERIMENTAL ANALYSIS

**Ablations on Backbone Initialization.**    We initialize our backbone network with MASt3R (Leroy et al., 2024) pre-trained weights in our main experiment. MASt3R is trained on datasets with ground truth depth annotation, but notably, its training data has no overlap with the training and evaluation datasets used in our experiments. Here, we also investigate the impact of different backbone initialization methods. Specifically, we compare the performance of our method using pre-trained weights from DUSt3R (Wang et al., 2024b), CroCo V2 (Weinzaepfel et al., 2023), and DINOv2 (Oquab et al., 2023) (DINOv2 is only used to initialize the encoder, the decoder is randomly initialized), as well as randomly initialized weights. DUSt3R is also trained with depth supervision but without feature matching loss compared with MASt3R. CroCoV2 is pre-trained with pure 2D image pairs with reconstruction loss (He et al., 2022), while DINOv2 (Oquab et al., 2023) is a self-supervised method trained on 2D images. Note that for CroCoV2, DINOv2, and random initialization, we warm up the training by adding point cloud distillation loss from the DUSt3R model for 1,000 steps, this aims to tell the network the goal is to predict the Gaussians in the canonical space, otherwise, the learning target is too hard for the network to understand as we only train our network with photometric loss. These results in Tab. 7 demonstrate that our method effectively learns pose-free inference capabilities during training, with appropriate weight initialization further enhancing performance. Notably, even with random initialization, our method significantly outperforms the pose-free baseline (CoPoNeRF). Moreover, utilizing CroCov2 and DINOv2 pre-trained weights enables our method to surpass the previous SOTA pose-required method (MVSplat), despite neither initialization method providing prior knowledge of pose or depth information.

**Ablations on Pose Condition.** To further demonstrate the effectiveness of our method in inferring geometry without pose information, we conducted an ablation study by incorporating additional pose conditioning into our network. Keeping all other factors constant, we converted the pose information into Plucker ray representations and concatenated them with the RGB images as network input. The experimental results, shown in Tab. 8, reveal that while pose conditioning slightly improves performance compared to our pose-free method, the enhancement is small especially when initialized with MASt3R weights. This highlights our method's robust ability to correlate image pairs even in the absence of explicit pose information.

**Ablations on the Effectiveness of Two-Stage Pose Estimation.**    As shown in Tab. 9, relying solely on PnP-RANSAC results in inaccurate pose estimates. However, if we skip the coarse camera

Table 9: **Ablation on the pose estimation method**.

| PnP | Photometric | 5° ↑ | 10° ↑ | 20° ↑ |
|:---:|:---:|:---:|:---:|:---:|
| ✓ | ✓ | **0.318** | **0.538** | **0.717** |
| ✓ | | 0.287 | 0.506 | 0.692 |
| | ✓ | 0.017 | 0.027 | 0.051 |

Table 10: **Performance comparison on the evaluation set of pixelSplat.**

| | PSNR↑ | SSIM↑ | LPIPS↓ |
|:---|:---:|:---:|:---:|
| pixelNeRF | 20.43 | 0.589 | 0.55 |
| GPNR | 24.11 | 0.793 | 0.255 |
| AttnRend | 24.78 | 0.82 | 0.213 |
| pixelSplat | 26.09 | 0.863 | 0.136 |
| MVSplat | 26.39 | 0.869 | 0.128 |
| Ours | **26.786** | **0.878** | **0.124** |

pose estimation stage and only use photometric loss-based optimization starting from $[\boldsymbol{I} \mid \boldsymbol{0}]$, the performance significantly degrades when using the same number of optimization steps (200 steps). This is because optimizing from an initial pose far from the target is more challenging.

**Evaluate on the Evaluation Set of pixelSplat.** We also present results based on the evaluation set used by pixelSplat (Charatan et al., 2024) and MVSplat (Chen et al., 2024). The results are shown in Tab. 10. However, we do not prioritize this evaluation set as it is relatively simple, with most input pairs exhibiting significant overlap, making it less suitable for more advanced future research.

**Details on Extension to 3 Input Views.** Our method can be extended to an arbitrary number of input views. As shown in Tab. 5, using three input views improves the results. Specifically, we add the middle frame between the original two input frames as an additional view. For three-input settings, when generating the Gaussians of each view, we concatenate the feature tokens from all other views and perform cross-attention with these concatenated tokens in the ViT decoder stage, keeping all other operations the same as with two input views. Fig. 13 provides a qualitative comparison between using two and three input views.

**Experiments with More Input Views.** We further evaluate the performance with varying numbers of input views. We train separate models for each configuration. The quantitative results are presented in Fig. 10. To ensure fair comparison, we maintain consistent first and last views across all experiments, while sampling intermediate views as the number of input views increases. The results demonstrate significant performance improvements when increasing the input views from two to three or four. However, additional input views beyond four yield comparable performance. This is because four views are sufficient to capture most of the scene information. This finding underscores our method's robust capability to effectively process sparse input images. Figure 11 provides qualitative examples across different view configurations, corroborating the quantitative results.

**Geometry Comparison with Splatt3R and MASt3R.** In Figure 5, we compare our generated 3D Gaussians with state-of-the-art pose-required feed-forward 3D Gaussian Splatting methods, namely pixelSplat and MVSplat. Additionally, we compare the geometric representations with MASt3R and Splatt3R. Specifically, we compare our 3D Gaussians with Splatt3R's 3D Gaussians and MASt3R's point cloud, as MASt3R only outputs point cloud representations. Notably, our method is trained solely using photometric loss, without ground truth point cloud supervision, whereas MASt3R utilizes ground truth point cloud for training, and Splatt3R directly employs MASt3R's point cloud outputs as Gaussian centers. The results in the first row of Figure 12 demonstrate that our method achieves better geometric details (*e.g.*, straighter wall edges) despite the absence of point cloud supervision. The second row reveals that our method more coherently fuses the content of two input images, while significant quality discontinuities exist in the intersection areas of MASt3R and Splatt3R. Furthermore, MASt3R fails to capture the geometric relationships of distant objects, resulting in severe blurring in these regions when Splatt3R directly uses its point cloud as Gaussian centers.

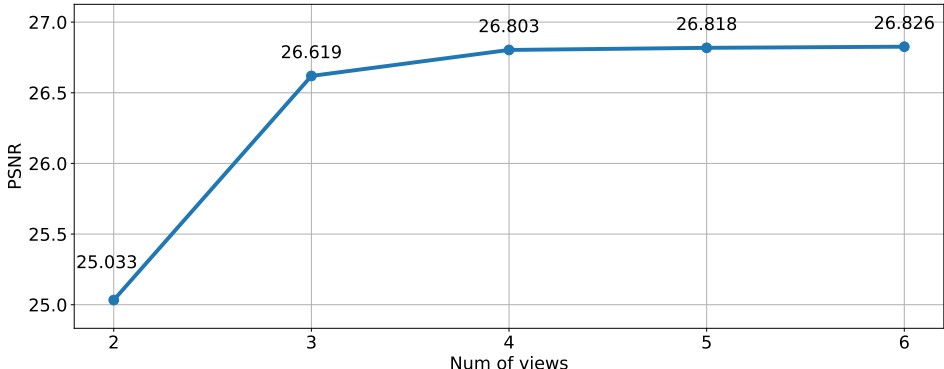

Figure 10: **Novel view synthesis results with different number of input views.**

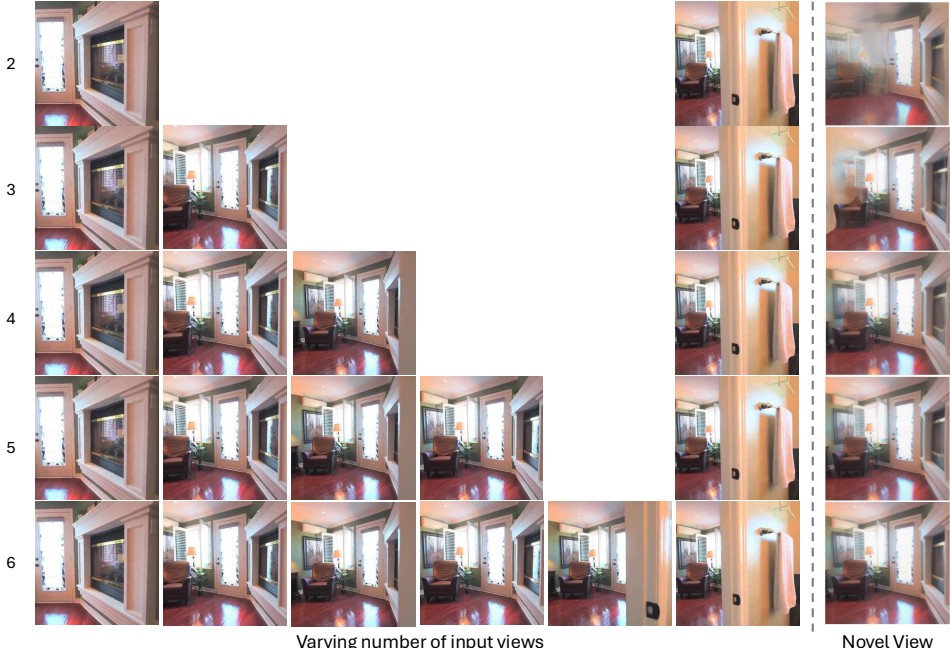

Varying number of input views         Novel View

Figure 11: **RealEstate10k performance with different number of input views.**

**Addtional Comparison with Splatt3R.** In Tab.1 of the main paper, we compare our method with the official model provided by the authors of Splatt3R(Smart et al., 2024), which freezes the MASt3R model and is then trained on the ScanNet++ (Yeshwanth et al., 2023) dataset. Here, we also train it on the RealEstate10K (Zhou et al., 2018) dataset at the same resolution as other baselines ($256 \times 256$). However, when attempting to retrain it using the official code provided by the authors on RealEstate10k, we find that the training fails because the original Splatt3R can only be trained on datasets with metric pose ground truth. This limitation arises because Splatt3R relies on a fixed pre-trained MASt3R (Leroy et al., 2024) model and its capability for metric depth prediction. As MASt3R is trained on ScanNet++ (Yeshwanth et al., 2023) using ground truth metric depth information, the poses in ScanNet++ are also metric. Consequently, the original Splatt3r cannot be trained on video datasets without metric pose information. We identify two main issues:

1. The camera poses provided by RealEstate10K are up-to-scale, as they are estimated using COLMAP (Schonberger & Frahm, 2016). This results in a scale misalignment between the ground truth pose and the scale of the estimated Gaussian field. Although Splatt3r predicts additional point offsets based on the point cloud estimated by MASt3R as the final Gaussian center, these offsets are typically small and unable to resolve the misalignment problem. This issue persists even in datasets with metric camera poses, as the point cloud provided by MASt3R is imperfect and not fully aligned with the ground truth pose scale. The training fails if the provided COLMAP target camera poses are used to render target views for the 3D Gaussians.

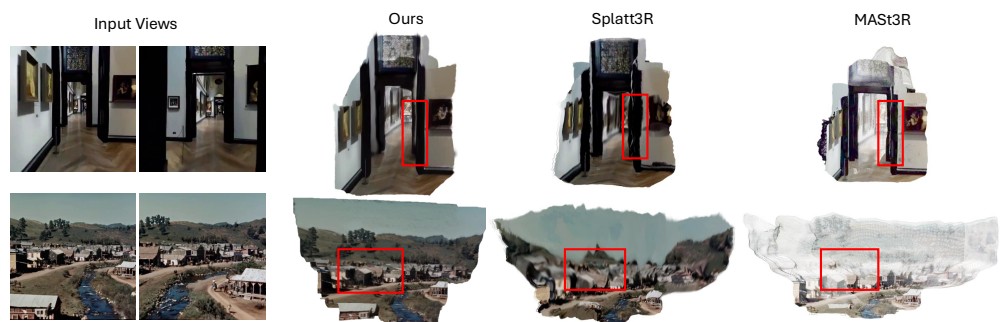

Figure 12: **Geometry comparison with Splatt3R and MASt3R.**

Table 11: **Performance of retrained Splatt3R (Smart et al., 2024) model.**

| | Small | | | Medium | | | Large | | | Average | | |
|---|---|---|---|---|---|---|---|---|---|---|---|---|
| | PSNR↑ | SSIM↑ | LPIPS↓ | PSNR↑ | SSIM↑ | LPIPS↓ | PSNR↑ | SSIM↑ | LPIPS↓ | PSNR↑ | SSIM↑ | LPIPS↓ |
| Ours | **22.514** | **0.784** | **0.210** | **24.899** | **0.839** | **0.160** | **27.411** | **0.883** | **0.119** | **25.033** | **0.838** | **0.160** |
| Splatt3R Official | 14.352 | 0.475 | 0.472 | 15.529 | 0.502 | 0.425 | 15.817 | 0.483 | 0.421 | 15.318 | 0.490 | 0.436 |
| Splatt3R Retrain | 17.987 | 0.616 | 0.385 | 19.362 | 0.657 | 0.327 | 20.518 | 0.685 | 0.288 | 19.354 | 0.655 | 0.330 |

To address this, we first estimate the camera poses of two input views using the point cloud from MASt3R, which is scale-consistent with the Gaussians. We then align the scale of the COLMAP target poses with the scale of the MASt3R point cloud by adjusting the scale of the input view poses to match the MASt3R-estimated ones.

2. We also find that the intrinsic parameters estimated by MASt3R (Leroy et al., 2024) do not align well with the ground truth intrinsic parameters. Using the ground truth intrinsic during training also causes failure. As a result, we opt to use the intrinsics estimated by MASt3R.

Additionally, we ignore the loss mask used in the original Splatt3R as it requires ground truth depth to generate the mask, but the RealEstate10K dataset lacks ground truth depth. The mask is unnecessary for training on video datasets like RealEstate10K, since during training, we use intermediate frames between the two input frames as targets, and most of the image content in the target frames is well covered by the input frames. For a fair comparison during evaluation, we apply the same evaluation-time pose alignment technique used in our method. The results are presented in Tab. 11. Although retraining the model on the RealEstate10K dataset yields improved performance, it still significantly lags behind our approach. This performance gap can be attributed to the misalignment issues inherent in the fixed MASt3R model and the persistent scale ambiguity problem.

## D    LIMITATIONS

Our approach, like previous pose-free methods (Hong et al., 2024a; Fu et al., 2024), assumes known camera intrinsics. Although heuristically set intrinsic parameters prove effective for in-the-wild images, relaxing this requirement would enhance the robustness of real-world applications. Additionally, as our feed-forward model is non-generative, it lacks the ability to reconstruct unseen regions of a scene with detailed geometry and texture, as demonstrated in the results of the 2-view model shown in Fig. 13. This limitation could potentially be mitigated by incorporating additional input views, which would enhance scene coverage. Finally, the current training data (limited to RealEstate10K, ACID, and DL3DV) constrains the model's generalizability to diverse in-the-wild scenarios. Future work could explore training our model on large-scale, diverse indoor and outdoor datasets, leveraging our method's independence from ground-truth depth information during training.

## E    MORE VISUAL COMPARISONS

We present an additional comparison with previous SOTA pose-dependent and pose-free methods across various levels of image overlap: Fig.14 for small input image overlap, Fig.15 for medium input image overlap, and Fig.16 for large input image overlap. Furthermore, we provide additional comparisons on the ACID dataset (Liu et al., 2021) in Fig.17.

Ref.   Target View   Ours (3 Views)   Ours (2 Views)

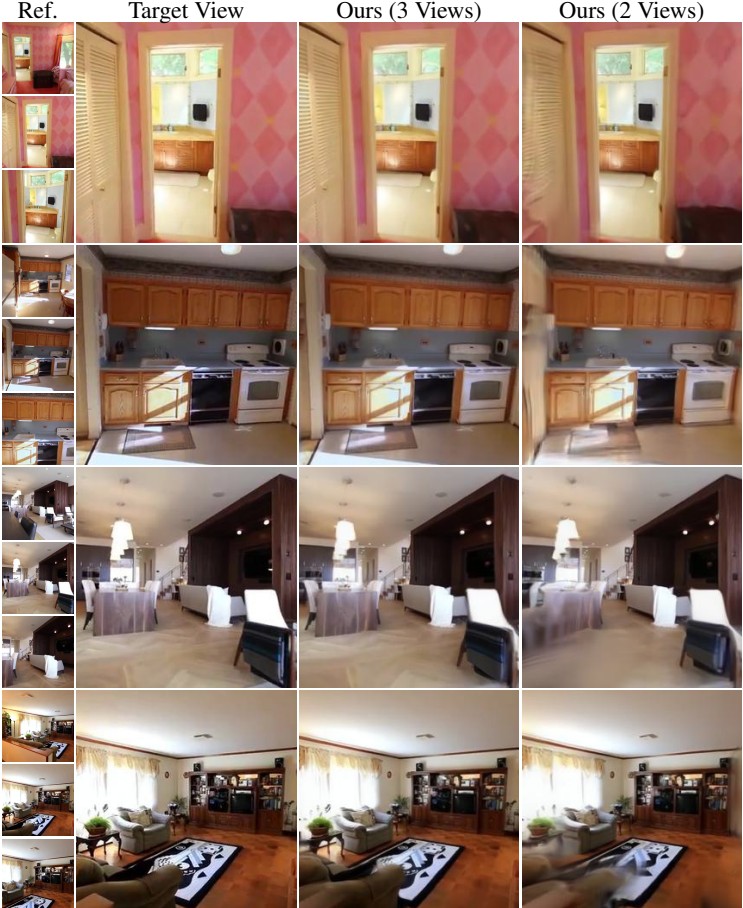

Figure 13: **Qualitative comparison on different numbers of input views.** The model with 2 input views utilizes only the top and bottom reference images, whereas the 3-view model incorporates an additional intermediate view. Adding this extra reference view improves the quality of rendered novel views significantly, as it captures finer spatial details and reduces occlusion effects.

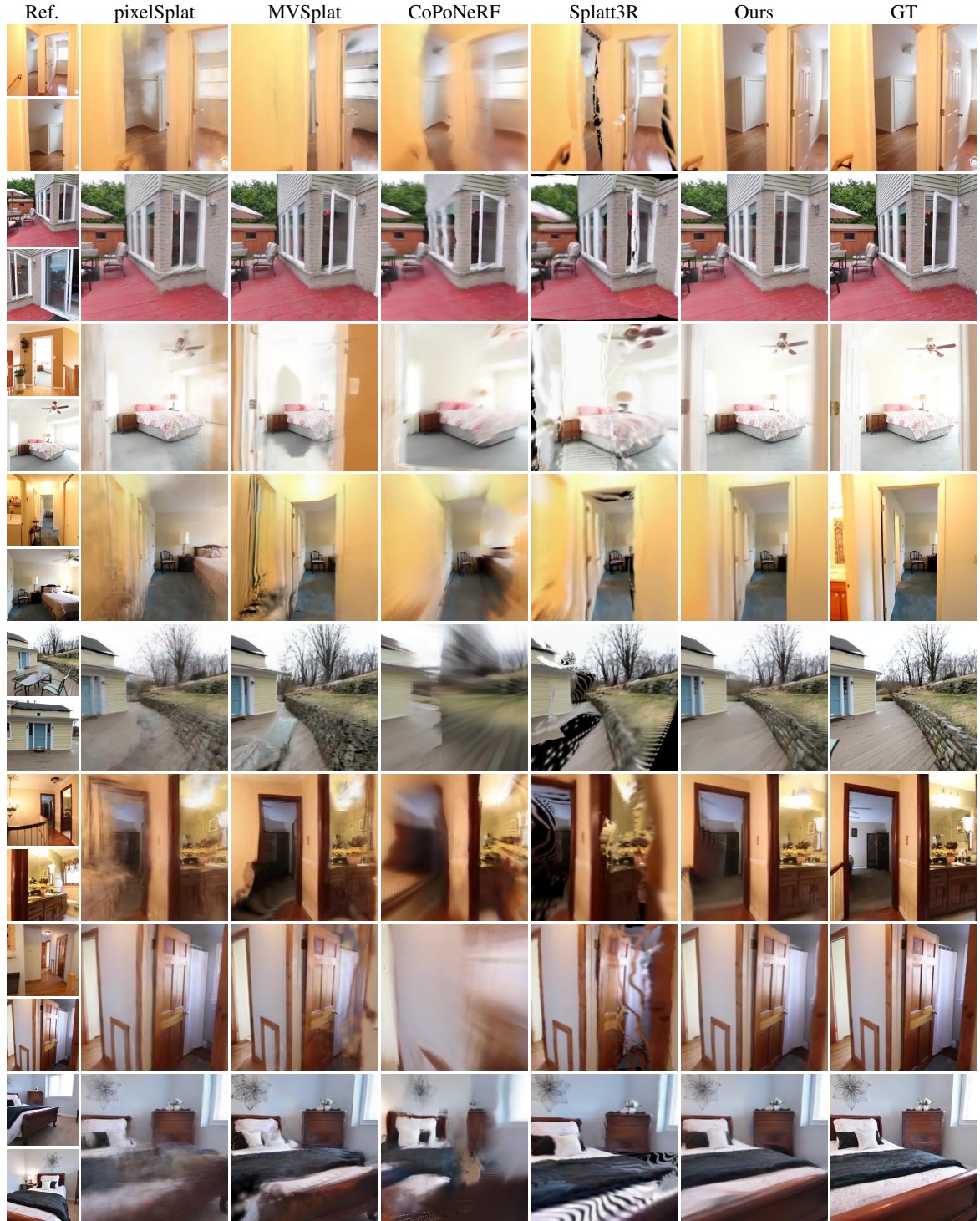

Figure 14: **More comparisons of the RealEstate10K dataset with** *small* **overlap of input images.**

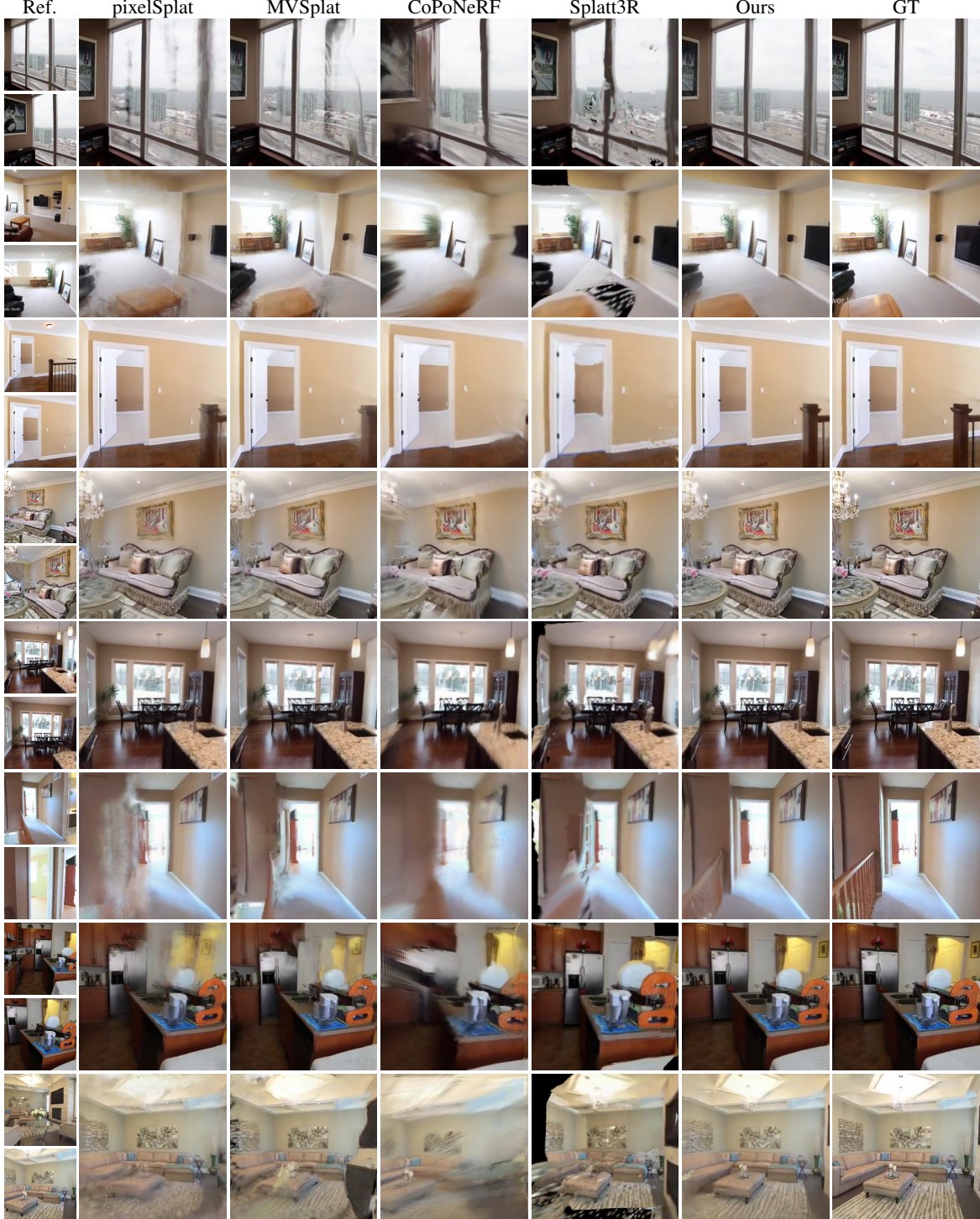

Figure 15: **More comparisons of the RealEstate10K dataset with *medium* overlap of input images.**

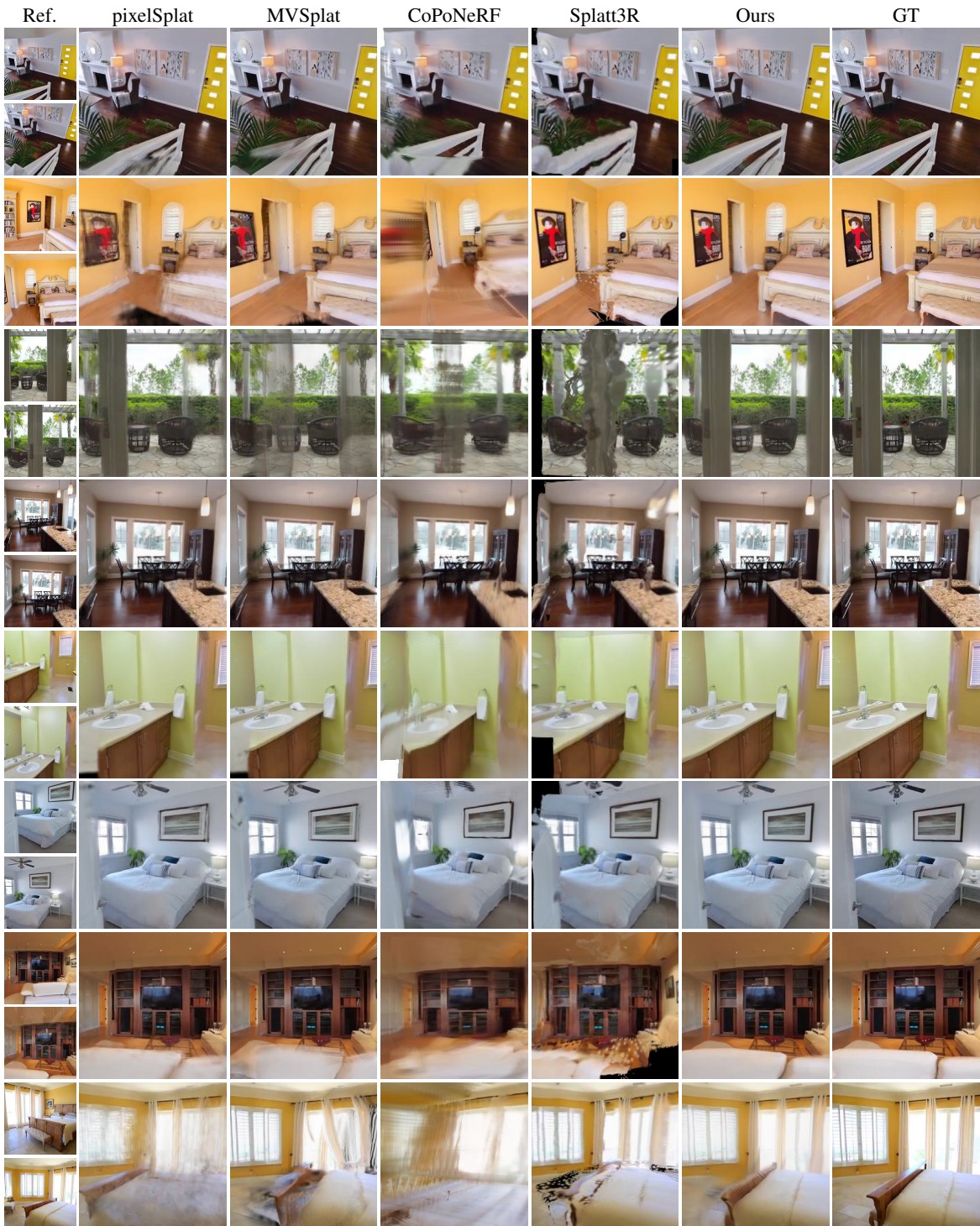

Figure 16: **More comparisons of the RealEstate10K dataset with *large* overlap of input images.**

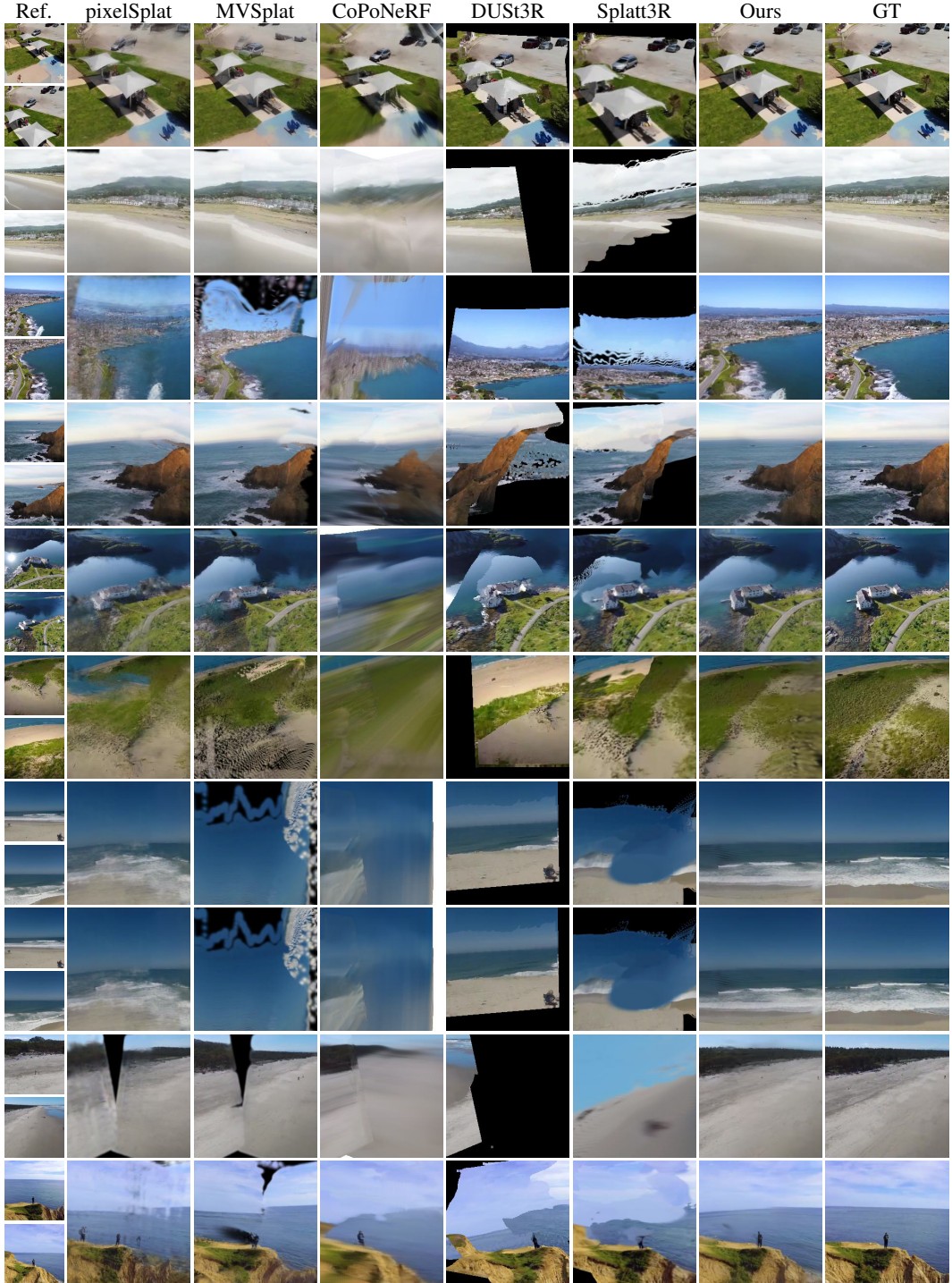

Figure 17: **More comparisons on the ACID dataset.**

