# OpenReview forum: "No Pose, No Problem: Surprisingly Simple 3D Gaussian Splats from Sparse Unposed Images"
_ICLR.cc/2025/Conference — ICLR 2025 Oral_

### Official Review · Reviewer_tg6H · 2024-10-26

**Soundness:** 3
**Presentation:** 2
**Contribution:** 1
**Rating:** 8
**Confidence:** 5

**Summary:**

This paper proposes a feed-forward generalizable GS method from sparse and unposed images. It encodes images (with known intrinsics) into feature tokens and use a decoder with cross-attention layers for cross-view feature fusion. Then the gaussians are predicted in a canonical space. The proposed method is evaluated on RealEstate10K/ACID and tested generalization on DTU.

**Strengths:**

- The paper proposes a simple and straightforward method.
- The paper is well-written and is easy to follow.
- The method shows a good performance, which is even better than previous pose-aware methods, e.g. MVSplat.

**Weaknesses:**

1. This paper is technically sound. However, basic literatures in this field are not discussed and the claimed novelty have already been studied in prior works. I don't understand why the authors ignore these works when the content and techniques are closely related. Please see details below.
- PF-LRM [1] and LEAP [2] are the two earliest works for pose-free reconstruction, but none of them are mentioned. Almost all claimed novelty of the submssion, e.g. canonicalization, PnP pose estimation, using cross-attention for cross-view feature fusion, have already been studied in these two works (also true for DUST3R). The only difference is whether using 3DGS or NeRF as the 3D representation (which is actually also studied in GS-LRM [3] and GRM [4], see below). However, I don't think this is a solid contribution for a publication.
- Besides, the architecture of the proposed work is almost the same with GS-LRM [3] with only two differences of design choice: (1) whether using a ViT encoder or directly tokenizing the images; (2) whether using cross-attention or using concatenation and self-attention for feature fusion. This is also true for GRM [4].

The paper is a direct combination of these previous works (pose-free feed-forward NVS from sparse-views + feed-forward 3DGS prediction).

- Moreover, methods that use latent 3D representations [5,6,7] (SRT is the first pose-free NVS method), reconstruction using sparse-view pose estimation methods [8,9,10,11] and more scene-specific/optimization-based pose-free methods [12] should be discussed.

[1] Wang, Peng et al. “PF-LRM: Pose-Free Large Reconstruction Model for Joint Pose and Shape Prediction.” ICLR 2024.
[2] Jiang, Hanwen et al. “LEAP: Liberate Sparse-view 3D Modeling from Camera Poses.” ICLR 2024.
[3] Zhang, Kai et al. “GS-LRM: Large Reconstruction Model for 3D Gaussian Splatting.” ECCV 2024.
[4] Xu, Yinghao et al. “GRM: Large Gaussian Reconstruction Model for Efficient 3D Reconstruction and Generation.” ECCV 2024.
[5] Sajjadi, Mehdi SM, et al. "Scene representation transformer: Geometry-free novel view synthesis through set-latent scene representations." CVPR 2022.
[6] Kani, Bharath Raj Nagoor, et al. "UpFusion: Novel View Diffusion from Unposed Sparse View Observations." ECCV 2024.
[7] Xu, Chao, et al. "Sparp: Fast 3d object reconstruction and pose estimation from sparse views." ECCV 2024.
[8] Zhang, Jason Y., Deva Ramanan, and Shubham Tulsiani. "Relpose: Predicting probabilistic relative rotation for single objects in the wild." ECCV 2022.
[9] Zhang, Jason Y., et al. "Cameras as rays: Pose estimation via ray diffusion." ICLR 2024.
[10] Jiang, Hanwen, et al. "Few-view object reconstruction with unknown categories and camera poses." 3DV 2024.
[11] Lin, Amy, et al. "Relpose++: Recovering 6d poses from sparse-view observations." 3DV 2024.
[12] Truong, Prune, et al. "Sparf: Neural radiance fields from sparse and noisy poses." CVPR 2023.

2. Moreover, there some other weaknesses as follows.
- Inappropriate baselines. DUST3R is not designed for NVS but for reconstruction with point cloud. Evaluating NVS performance on DUST3R is misleading. This is also the same for MAST3R/Splatt3R. If you want to compare with these works, please compare the geometry quality, i.e. point cloud error, rather than NVS performance. Moreover, DUST3R and its following works don't require camera intrinsics.
- Unfair comparison. The proposed method is trained on a joint set of RealEstate10K, DL3DV and ACID, while all the baselines are trained on a different training dataset. Moreover, the model uses the weight initialization of models trained on large dataset (MAST3R/DUST3R/Croco).

3. As there are not a lot directly comparable works on scene-level, I require the authors to include additional five experiment results.
- On object-level, train the proposed model using the same data with the previously mentioned pose-free works and compare their performance. This also verifies whether MAST3R weight initialization generalize to object-level data.
- On scene-level, train with only RealEstate10K+ACID for comparing with MVSplat.
- On scene-level, train a variant of the proposed model, which is also conditioned on the camera poses of inputs (using the plucker ray representation rather than 6D pose representation). If the performance gap is small enough, the model has a strong capability of correlating the two images with the missing poses.
- On scene-level, ablate the weight initialization by training with no weight initialization (from MAST3R/DUST3R/Croco) using the current training set (ACID+RealEstate10K+DL3DV). This experiment ablates whether the pose-free inference capability comes from initialization or your model learning process. Please include results for both pose-free and pose-conditioned variants of your model.

4. Besides, the authors should notify that during training, the method still requires camera poses for rendering.

5. Camera intrinsics are still needed during inference. The proposed way of merging camera intrinsic information is the same as LRM/PF-LRM, but no discussion.

6. The method is only tested with two views, while the author claim "sparse-view". The authors should perform experiments with more input views, e.g. up to 6-8 views, to support the claim. Otherwise, you should rename the paper as "unposed two images".

**Questions:**

Please first see comments above.

**It is necessary to discuss prior works in an appropriate way.** I will give a reject first but I can raise the score to 5 if the author can differentiate their work from the missing related works and rephrase their contribution.

**Besides, I strongly require the authors to perform experiment using the same training dataset with previous works.** For example, if you want to compare with MVSplat, you should only train on RealEstate10K+ACID. This is the correct way of doing scientific research. Otherwise, we cannot tell whether the performance gain comes from better model design or using larger and high-quality data (and also your MAST3R weight initialization should be ablated). The additionally used datasets in this paper are all from available resources and there is no need for data curation. It is different from LLM research where data collection and curation and be their contribution, and, thus, it is acceptable to directly compare LLMs trained using different data.

I can raise the score to 6 or higher scores if the authors performs the asked three experiments (weakness 3) with promising results.

---

> ### Author Response · Authors · 2024-11-23
> **Official Response by Authors -- Part 1**
>
> We would like to sincerely thank the reviewer for the useful comments on our work. To effectively respond to all your concerns, we first provide all the additional experiments requested by the reviewer, and then respond to all remaining comments.
>
> We take every comment seriously and hope our response can address the reviewer’s concerns. If there are any remaining questions, we are more than happy to address them.
>
> ## Additionally Requested Experiments
> > **1. On object-level, train the proposed model using the same data with the previously mentioned pose-free works and compare their performance. This also verifies whether MAST3R weight initialization generalizes to object-level data.**
>
> Thank you for your suggestion. We agree that such an experiment can further verify the generalizability of our method to object-level reconstruction tasks.
>
> The reviewer mentioned two pose-free object-level reconstruction works: PF-LRM and LEAP. Unfortunately, PF-LRM has not open-sourced its code or dataset, making a direct comparison infeasible. Therefore, we conducted a comparison with LEAP, the state-of-the-art open-source pose-free method for object-level reconstruction.
>
> To ensure a fair comparison, we trained and evaluated both methods on the widely adopted Objaverse dataset, maintaining identical training iterations and input image resolutions. The results, shown in the table below, demonstrate that our approach significantly outperforms LEAP, with the PSNR improving from 20.559 to 28.378. This highlights the robust generalization capabilities of our method for object-level reconstruction. The qualitative comparison is presented in Fig. 9 of our revised manuscript.
>
> |      |  PSNR  |  SSIM | LPIPS |
> |------|:------:|:-----:|:-----:|
> | LEAP | 20.559 | 0.853 | 0.144 |
> | Ours | 28.378 | 0.935 | 0.053 |
>
> ---
> > **2. On scene-level, train with only RealEstate10K+ACID for comparing with MVSplat.**
>
> We would like to clarify a misunderstanding regarding our experimental setup. As mentioned in **L311–313** of the paper, we strictly adhere to the experimental setup used by pixelSplat and MVSplat when comparing with these methods, without incorporating the DL3DV dataset. Specifically:
> - For the RealEstate10K dataset, we train our model exclusively on this dataset, following the exact experimental protocol employed by pixelSplat and MVSplat.
> - Similarly, for the ACID dataset, our model is trained solely on this dataset for the corresponding evaluation.
>
> The version of our model trained on both RealEstate10K and DL3DV is denoted as Our* throughout the paper (see L.315–317) and is not used for comparing novel view synthesis performance. This distinction ensures that all comparisons are fair and consistent with prior works.
>
> We hope this clarifies the concern.
>
> ---
> > **3. On scene-level, train a variant of the proposed model, which is also conditioned on the camera poses of inputs (using the plucker ray representation rather than 6D pose representation). If the performance gap is small enough, the model has a strong capability of correlating the two images with the missing poses.**
>
> Thank you for your suggestion. Following your recommendation, we incorporated pose information by converting it into Plücker ray representation and concatenating it with the RGB image as network input. Our results below demonstrate that while such pose conditioning slightly improves performance compared to our pose-free method, the enhancement is **marginal**. This indicates our method's robust capability in correlating image pairs even without explicit pose information.
>
>
> | Init   Weight | pose condition |  PSNR  |  SSIM | LPIPS |
> |:-------------:|:--------------:|:------:|:-----:|:-----:|
> |     MASt3R    |       Yes      | 25.080 | 0.844 | 0.158 |
> |               |       No       | 25.033 | 0.838 | 0.160 |
> |     Random    |       Yes      | 23.708 | 0.788 | 0.173 |
> |               |       No       | 23.487 | 0.779 | 0.189 |

---

> ### Author Response · Authors · 2024-11-23
> **Official Response by Authors -- Part 2**
>
> > **4. On scene-level, ablate the weight initialization by training with no weight initialization (from MAST3R/DUST3R/Croco) using the current training set (ACID+RealEstate10K+DL3DV). This experiment ablates whether the pose-free inference capability comes from initialization or your model learning process. Please include results for both pose-free and pose-conditioned variants of your model.**
>
> Thank you for the insightful suggestion.
>
> To address this request, we conducted additional experiments under several initialization settings:
> 1. *DINOv2 Initialization*: Since DINOv2 is only a ViT encoder, we used it to initialize our encoder, while the decoder was randomly initialized.
> 2. *Random Initialization*: We trained our model from entirely random weights, including both the encoder and decoder.
>
> The results indicate that our method effectively learns pose-free inference capabilities during training, regardless of initialization. However, appropriate weight initialization enhances performance further. Specifically:
> - Even with random initialization, our method significantly outperforms the pose-free baseline, CoPoNeRF.
> - Using pre-trained weights from CroCo-v2 or DINOv2 enables our method to surpass the previous performance of MVSplat, despite neither initialization method incorporating prior knowledge of pose or depth information.
>
> Additionally, it is worth noting that weight initialization is a standard practice in the community. For example: pixelSplat utilizes DINOv2 features for initialization. MVSplat uses Unimatch [1], which has been pre-trained on various depth datasets.
> Neither pixelSplat nor MVSplat employs random initialization, as proper weight initialization is critical for efficient convergence and improved performance.
>
> | Init Weights|  PSNR  |  SSIM | LPIPS |
> |-------------|:------:|:-----:|:-----:|
> | MASt3R      | 25.033 | 0.838 | 0.160 |
> | CroCo-v2    | 24.559 | 0.818 | 0.171 |
> | DINOv2      | 24.094 | 0.812 | 0.176 |
> | Random      | 23.487 | 0.779 | 0.189 |
>
> [1] Xu, Haofei, et al. "Unifying flow, stereo and depth estimation." TPAMI (2023).

---

> ### Author Response · Authors · 2024-11-23
> **Official Response by Authors -- Part 3**
>
> ## All Other Comments
>
> > **Q1: PF-LRM and LEAP are the two earliest works for pose-free reconstruction, but none of them are mentioned. Almost all claimed novelty of the submssion, e.g. canonicalization, PnP pose estimation, using cross-attention for cross-view feature fusion, have already been studied in these two works (also true for DUST3R). The only difference is whether using 3DGS or NeRF as the 3D representation (which is actually also studied in GS-LRM and GRM, see below). However, I don't think this is a solid contribution for a publication**
>
> Thank you for highlighting these relevant works. We acknowledge the contributions of PF-LRM, LEAP, and other pose-free methods to the field of object-level reconstruction. While these methods primarily focus on **object-level data**, our work is centered on **scene-level data**, addressing the unique challenges and opportunities it presents. We do not intend to disregard the significance of prior works, but rather, aim to expand the scope and capabilities of pose-free reconstruction methods.
>
> **Key differences and contributions of our work include:**
> - *Efficiency and Scalability with 3DGS*: PF-LRM and LEAP employ NeRF as a representation, but their reliance on volumetric rendering leads to low efficiency and resolution constraints, which limit their applicability for complex scene-level reconstruction. In contrast, our work focuses on developing a simple yet efficient pipeline for pose-free, feedforward Gaussian prediction. While feedforward Gaussian prediction has been explored previously, adapting it to a pose-free, scene-level setting with explicit representation is non-trivial, and we address these challenges in our work.
> - *Canonical Gaussian Prediction as a Novel Exploration*: While previous methods like pixelSplat and MVSplat rely on per-view local Gaussian prediction, we explore the advantages of canonical prediction, which enables more coherent feature fusion across views (see Fig. 5 and row f in Tab. 5). Canonical Gaussian prediction in the pose-free setting remains underexplored in the existing literature and is a core novelty of our approach.
> - *Accurate Scene-level Pose Estimation*: Unlike PF-LRM and LEAP, which focus on novel view synthesis and provide limited pose estimation results on object-level data, our method demonstrates superior performance in pose estimation on complex indoor and outdoor scenes (see Tab. 3).
> - *Innovations in Network Design*: We introduce an RGB shortcut in the network (see Sec. 3.2 and row e in Table 5) to enhance the prediction of fine texture details, which improves rendering quality, a contribution not present in PF-LRM and LEAP.
> - *Investigation in Intrinsic Embeddings*: While PF-LRM also explores intrinsic embedding, we investigate its necessity and optimal implementation for generalizable Gaussian Splatting prediction, which is critical for novel view synthesis.
>
> ---
> > **Q2: The architecture of the proposed work is almost the same with GS-LRM with only two differences of design choice: (1) whether using a ViT encoder or directly tokenizing the images; (2) whether using cross-attention or using concatenation and self-attention for feature fusion. This is also true for GRM.**
>
> We would like to clarify that **network architecture is not the primary focus or contribution** of our work, nor did we claim it as a novelty. As indicated in our title, we employ a simple and standard Vision Transformer-based architecture. The focus of our work lies in 1) reconstructing 3D Gaussians from unposed images efficiently and accurately, and 2) investigating how canonical Gaussian prediction benefits both novel view synthesis and pose estimation tasks, particularly in scene-level data.
>
> Our method is not a direct combination of previous pose-free NVS work like PF-LRM, and LEAP. Instead, we address the underexplored design of pose-free, feedforward 3DGS pipelines and demonstrate the previously unknown advantages of canonical Gaussian prediction over per-view methods like MVSplat.
>
> > **Q3: methods that use latent 3D representations (SRT is the first pose-free NVS method), reconstruction using sparse-view pose estimation methods and more scene-specific/optimization-based pose-free methods should be discussed.**
>
> Thank you for all the suggested papers. We have added discussions to all papers in Sec. 2 in the revised manuscript.

---

> ### Author Response · Authors · 2024-11-23
> **Official Response by Authors -- Part 4**
>
> > **Q4: Inappropriate baselines. DUST3R is not designed for NVS but for reconstruction with point cloud. Evaluating NVS performance on DUST3R is misleading. This is also the same for MAST3R/Splatt3R. If you want to compare with these works, please compare the geometry quality, i.e. point cloud error, rather than NVS performance.**
>
> We included DUSt3R and MASt3R in our novel view synthesis (NVS) comparisons following Splatt3R, because as noted also by other reviewers, Splatt3R is a closely related approach trained for NVS and it shows the comparison to these two methods. To ensure consistency and facilitate a fair comparison, we followed the same evaluation protocols as Splatt3R to include all three methods and included them in our comparison.
>
> That said, if the reviewer feels that including these methods in the comparison is not appropriate given their original focus, we are happy to revise the manuscript and remove these comparisons in the final version.
>
> **Comparing the geometry quality:** We would like to clarify first that MASt3R/DUSt3R focuses primarily on point-matching quality and pose estimation, while Splatt3R evaluates performance only through novel view synthesis experiments. Nevertheless, we have included a qualitative comparison of geometry reconstruction by visualizing 3D Gaussians or point clouds in our revised manuscript, which can be found in Fig. 12 and L.955–967. The results demonstrate that our approach captures finer geometric details and higher geometry quality than both MASt3R and Splatt3R.
>
> This outcome is particularly encouraging, as our method obtains such high-quality geometry using only photometric loss as supervision during training, while both MASt3R and Splatt3R rely on ground-truth point cloud supervision.
>
> ---
>
> > **Q5: Unfair comparison. The proposed method is trained on a joint set of RealEstate10K, DL3DV and ACID, while all the baselines are trained on a different training dataset. Moreover, the model uses the weight initialization of models trained on large dataset (MAST3R/DUST3R/Croco).**
>
> As detailed in our response to the first additional experiment, the comparison is fair since we strictly adhere to the experimental setup of pixelSplat and MVSplat when comparing with these methods.
> Specifically, we train and evaluate our model on the same datasets as these baselines without incorporating the joint set for these specific comparisons. Please refer to the first additional experiment response for more details.
> Regarding weight initialization, this is a standard technique within the deep learning community. Neither pixelSplat nor MVSplat utilize random initialization. Specifically, pixelSplat employs DINO feature initialization, while MVSplat is initialized using Unimatch [1], which has been pre-trained on various depth datasets.
>
> > **Q6: The authors should notify that during training, the method still requires camera poses for rendering.**
>
> Thank you for your suggestion, we have added this information in Sec. 3.5.
>
> ---
> > **Q7: Camera intrinsics are still needed during inference. The proposed way of merging camera intrinsic information is the same as LRM/PF-LRM, but no discussion.**
>
> Camera intrinsics are commonly required in previous pose-free methods such as CoPoNeRF and PF-LRM. However, as shown in row (b) of Tab.5, even without camera intrinsics, our method still shows comparable performance to the SOTA pose-required intrinsic-required method MVSplat and largely outperforms the SOTA pose-free intrinsic-required method CoPoNeRF.
>
> While intrinsic embedding is not a novel concept (as also utilized in PF-LRM), its necessity for generalizable Gaussian Splatting prediction and the optimal embedding approach—among several methods we compare—have not been thoroughly explored. We have included this discussion in Sec. 3.4 of the revised manuscript.
>
> ---
>
> > **Q8: method is only tested with two views, while the author claim "sparse-view". The authors should perform experiments with more input views, e.g. up to 6-8 views, to support the claim. Otherwise, you should rename the paper as "unposed two images".**
>
> While the two-view setting is a primary focus of the paper, we have also explored scenarios with more input views:
> - *Quantitative Results*: Tab. 5 includes results for three input views
> - *Qualitative Results*: We conducted additional experiments with up to 6 input views, see Fig. 10.

---

> > ### Comment · Reviewer_tg6H · 2024-11-23
> >
> > Thanks for the detailed feedback from the authors.
> >
> > The rebuttal solved most of my concerns. The reformulated contributions of this paper, discussion of prior methods, additional experiments on object-level, ablation with pose condition, clarification of training settings, and extension to more input views make this paper sound enough. Now I can see this paper is very ready to be published.
> >
> > I also encourage the authors to include experiments of initialization to the main paper.
> >
> > I will change my score to 8.

---

> > > ### Author Response · Authors · 2024-11-23
> > >
> > > Thank you for your response! We’re delighted to hear that our reply addressed your concerns. We truly appreciate your thorough and valuable feedback, which has greatly helped improve our work.
> > >
> > > Regarding your suggestion, due to the page limit, we have currently included the initialization experiments in the appendix. However, we will make an effort to reorganize the paper and incorporate these experiments into the main paper in our final version.

---

### Official Review · Reviewer_wkJy · 2024-10-29

**Soundness:** 3
**Presentation:** 3
**Contribution:** 3
**Rating:** 8
**Confidence:** 4

**Summary:**

The paper proposes a fast feed-forward model for sparse-view 3D reconstruction that is generalizable across scenes. Differently from recent work, the predictions (3DGS parameters) are performed in a common coordinate space directly without requiring any pose info. They perform evaluations on the RealEstate10k and ACID datasets and show improved performance over prior works. Their method also shows zero-shot generalizability to other datasets and in-the-wild scenarios. They also show relative pose estimation using a two-step approach: first with pnp and then with photometric optimization. The performance is benchmarked on similar datasets as well.

**Strengths:**

The proposed method is simple and easy to understand. More specifically, the highlighted advantages are:
- A simple VIT approach is competitive to pretrained ViT backbones from Dust3r, and Mast3r when overlap is minimal.
- The method does not require any ground-truth depth (explicit matching loss) during training thus allowing it to scale further. This is an important advantage since simple architectures w/ large datasets is a recipe for improving generalizability.
- Using pretrained Mast3r weights with pure photometric losses on more datasets is a good strategy to scale up training and improve model performance on nvs and relative pose estimation.

**Weaknesses:**

The canonical space motivation: The central argument of the paper is that predicting the scene structure in a common coordinate space improves scene geometry and multi-view cohesion compared to local prediction->global fused prediction approach. Although I agree with this statement, the paper does not exploit this advantage:
  * Given two views (which is the only input setting shown in the paper), the method still predicts per-pixel per-view 3DGS fields in a common coordinate (similar to Splatt3r). This is similar to Dust3r pointcloud variants where the pointclouds are predicted in a common coordinate space. I fail to see how this part is different from Dust3r-style prediction. Can you explain how this approach differs from those in the 2-view setting, if at all?
  * The main advantage seems to come from finetuning the Mast3r backbone on more data using photometric loss alone. By directly predicting 3DGS fields, this approach can improve overall model performance. But this orthogonal to the motivation of the method. It would be good to highlight this as one of the main features of this method

This approach is quite similar to Splatt3r although it is concurrent work. The advantage of this approach is that it does not need ground-truth depth during training which is important.

**Questions:**

- Contribution no. 4 is not a contribution.
- Multi-view (N>2): Are there any qualitative results for input images more than 2? What would the upper limit on this be given compute constraints? How does the cross-attention in the ViT decoder work for this case since L215 mentions "features from each view interact with those from all other views.."? Also, adding qualitative and quantiative results for N>2 and performance tradeoff cuves would strengthen this method.
- L274: "By default, we utilize the intrinsic token option". This needs to be "Global Intrinsic Embedding - Concat option".
- Table 1, Fig4: Why is Dust3R and Mast3r part of the NVS comparison since they were never trained for novel view synthesis? What was the evaluation protocol for these methods?
- In Fig4, its possible that the Gaussian structure masks the actual geometry of the scene when comparing to pointcloud outputs of Dust3r. To show geometry, it would be better to either show 3D points or depth maps. To show NVS, it would be better to remove Dust3r output since it is not a fair comparison.
- Discrepancy in metrics:
  * Both MVSplat and pixelSplat report higher PSNR numbers for RealEstate10k and ACID datasets. Since the same train-test split was used, why is there a huge discrepancy in the 'Average' column of the metrics?
  * Table 4: why might be the Splatt3r numbers lower than those reported in the original paper?
- Mentioning the training time and compute requirements would be useful additions to the paper.

Appendix
- Tab. 6 ablation does not provide the complete picture. The NVS metrics are dependent on the accuracy of the 3DGS field. Since the 3DGS parameter head is randomly initialized irrespective of the backbone, its contribution due to backbone initialization might not be as pronounced. A more accurate ablation would be on pose estimation given pure 3D points.
- L773, L775 typo. "experiences" -> experiments

**Details Of Ethics Concerns:**

No ethical concerns.

---

> ### Author Response · Authors · 2024-11-23
> **Official Response by Authors -- Part 1**
>
> We would like to sincerely thank the reviewer for the detailed and insightful comments on our work. We take every comment seriously and hope our response can address the reviewer’s concerns. If there are any remaining questions, we are more than happy to address them.
>
> > **Q1: Given two views (which is the only input setting shown in the paper), the method still predicts per-pixel per-view 3DGS fields in a common coordinate (similar to Splatt3r). This is similar to Dust3r pointcloud variants where the pointclouds are predicted in a common coordinate space. I fail to see how this part is different from Dust3r-style prediction. Can you explain how this approach differs from those in the 2-view setting, if at all?**
>
> The key difference lies in the **scene representation** itself. In our method, we predict 3DGS fields in a canonical coordinate space, whereas DUSt3R/MASt3R predicts a global point map. This distinction fundamentally impacts the focus and applications of these methods:
> - DUSt3R/MASt3R is designed to output point maps for downstream tasks such as point matching and pose estimation.
> - In contrast, our work investigates how canonical Gaussian prediction can advance novel view synthesis from unposed images, making it a distinct focus and contribution in this domain.
>
> Despite this difference in focus, our method achieves competitive results in other areas as well. Remarkably, although our approach does not involve training with ground-truth point maps, it produces a more detailed and geometrically correct 3D representation compared to MASt3R, see Fig. 12 in the revised paper. Additionally, our method also outperforms MASt3R/DUSt3R in relative pose estimation across many datasets.
>
> ---
> > **Q2: The main advantage seems to come from finetuning the Mast3r backbone on more data using photometric loss alone. By directly predicting 3DGS fields, this approach can improve overall model performance. But this orthogonal to the motivation of the method. It would be good to highlight this as one of the main features of this method.**
>
> Thank you for your insightful comment.
> 1. **We agree with the reviewer** that using photometric loss alone to fine-tune a ViT model for predicting canonical 3DGS significantly improves overall model performance compared to methods like MASt3R and DUSt3R. This extension offers several notable benefits:
>     - High-quality novel view synthesis
>     - More accurate pose estimation
>     - Broader dataset applicability (we can train purely on posed RGB image sequences or videos without GT point maps provided)
> 2. At the same time, we would like to note that our motivation in the paper is still valid, from the perspective of feed-forward sparse-view 3DGS methods. Our goal there is to address key limitations of previous approaches, such as the dependency on ground-truth poses and the misalignment issues in the 3DGS space. By directly predicting canonical 3DGS fields, we resolve these constraints.
>
> **Future Revisions**
> We strongly agree with the reviewer that the first point represents an important orthogonal feature of our method. To address this, we will refine the motivation in our final version to incorporate both perspectives: the benefits of fine-tuning with photometric loss and the advantages of predicting canonical 3DGS fields for overcoming prior limitations.

---

> ### Author Response · Authors · 2024-11-23
> **Official Response by Authors -- Part 2**
>
> > **Q3: This approach is quite similar to Splatt3r although it is concurrent work. The advantage of this approach is that it does not need ground-truth depth during training which is important.**
>
> We agree with the reviewer that a key distinction of our method compared to Splatt3R is that we do not require ground-truth depths or point cloud supervision during training. This is a significant advantage, as it enables our approach to leverage a broader range of easily obtainable video datasets, such as RE10K and DL3DV, which enhances performance.
>
> Beyond this, as we pointed out in the response to reviewer hSJP, our method also introduces several additional differences and improvements:
>
> - *End-to-End Training for Enhanced Performance*
> Another key difference is in training strategy. Unlike Splatt3R, which freezes the pretrained weights of MASt3R and trains only the Gaussian head, our method trains all network parameters end-to-end. This design choice addresses potential misalignments in the point clouds predicted by MASt3R, leading to more consistent renderings (see Fig. 4 in the paper). Additionally, this strategy supports varying numbers of input views (e.g., three views, as shown in Tab. 5), further demonstrating its robustness.
> - *RGB Image Shortcut for Improved Detail Capture*
> We also incorporate an RGB image shortcut in the Gaussian head (L.224–227), which allows our method to better capture fine texture details for Gaussian prediction, see row (e) of Tab. 5.
> - *Intrinsic Injection for Scale Ambiguity Resolution*
> As detailed in Sec. 3.4, we also include an intrinsic injection module to explicitly address the scale ambiguity problem and further boost performance.
>
> ---
> > **Q4: Contribution no. 4 is not a contribution.**
>
> Thank you for the suggestion. We have revised the manuscript accordingly and removed contribution 4 as a standalone contribution.
>
> Nevertheless, we would like to note that while we have removed this point from the contributions list, we believe these results remain valuable for showcasing the broader applicability and effectiveness of our approach.
>
> ---
> > **Q5: Multi-view (N>2): Are there any qualitative results for input images more than 2? What would the upper limit on this be given compute constraints? How does the cross-attention in the ViT decoder work for this case since L215 mentions "features from each view interact with those from all other views.."? Also, adding qualitative and quantiative results for N>2 and performance tradeoff cuves would strengthen this method.**
>
> Thank you for your thoughtful suggestions. While the two-view setting is a primary focus of the paper, we have also explored scenarios with more input views:
> - *Quantitative Results*: Tab. 5 includes results for three input views
> - *Qualitative Results*: We conducted additional experiments with up to 6 input views, see Fig. 11.
> - *Performance vs. Input Views*: We also provide a performance curve in Fig. 10, which illustrates how our method scales with additional views.
>
> Other questions:
> - *Upper limit of view number*: The maximum input number of views is 46 on a 24GB NVIDIA RTX4090 GPU.
> - *How does the cross-attention work*: When the number of views is greater than two, to generate the Gaussians for view $i$, we concatenate the features of all other views except $i$ and use these concatenated features as the keys and values in the cross-attention operation.
>
>
> ---
> > **Q6: Table 1, Fig4: Why is Dust3R and Mast3R part of the NVS comparison since they were never trained for novel view synthesis? What was the evaluation protocol for these methods?**
>
> We included DUSt3R and MASt3R in our novel view synthesis (NVS) comparisons because, as you also noted, Splatt3R—a closely related approach trained for NVS—conducted NVS comparisons with these two methods. To ensure consistency and facilitate a fair comparison, we followed the same evaluation protocols as Splatt3R for these two methods and included them in our comparison.
>
> That said, if the reviewer feels that including DUSt3R and MASt3R in the comparison is not appropriate given their original focus, we are happy to revise the manuscript and remove these comparisons in the final version.

---

> ### Author Response · Authors · 2024-11-23
> **Official Response by Authors -- Part 3**
>
> > **Q7: In Fig 4, its possible that the Gaussian structure masks the actual geometry of the scene when comparing to pointcloud outputs of Dust3r. To show geometry, it would be better to either show 3D points or depth maps. To show NVS, it would be better to remove Dust3r output since it is not a fair comparison.**
>
> 1. Similar to our response in Q6, we followed the same evaluation protocol as Splatt3R to assess the NVS performance of DUSt3R. We add them for consistency with the evaluation with prior work. However, if the reviewer feels it is not a fair comparison, we are happy to remove DUSt3R/MASt3R from the final version.
> 2. For geometry reconstruction, we agree that directly showing 3D points or depth maps could better illustrate the underlying scene geometry. To address this, we have added a comparison with MASt3R in Fig. 12 of the revised manuscript.
>
> ---
>
> > **Q8: Both MVSplat and pixelSplat report higher PSNR numbers for RealEstate10k and ACID datasets. Since the same train-test split was used, why is there a huge discrepancy in the 'Average' column of the metrics?**
>
> 1. While we use the same train-test split as MVSplat and pixelSplat, the discrepancy in the 'Average' column arises from differences in the image pairs used for evaluation. Specifically, the image pairs in their evaluation have large overlaps, which simplifies the task and makes it difficult to distinguish the true capabilities of each method in novel view synthesis NVS.
> 2. To better assess each method’s ability to handle varying degrees of camera overlap, we generate evaluation input pairs categorized by their overlap ratios: small (5%–30%), medium (30%–55%), and large (55%–80%). This categorization ensures a more rigorous evaluation, as described in L.319–L.321 of the paper.
> 3. The "Average" column in our results corresponds to the average performance across these three overlap settings. This explains why the performance appears lower compared to evaluations focused on large-overlap pairs.
> 4. To address any potential concerns, we have also reported results using the original evaluation set from pixelSplat and MVSplat in Tab. 8. Even under their evaluation settings, our method outperforms both pixelSplat and MVSplat, which require poses as input.
>
> ---
>
> > **Q9: Table 4: why might be the Splatt3r numbers lower than those reported in the original paper?**
>
> Similar to Q8, the discrepancy in Splatt3R's numbers arises from differences in the evaluation protocols. Specifically, we have chosen more varied overlap ratios, thus making the evaluation more challenging. As a result, the numbers reported for Splatt3R in Tab. 4 differ from those in their original paper.
>
> ---
> > **Q10: Mentioning the training time and compute requirements would be useful additions to the paper.**
>
> For the *256×256* version of the model, training was conducted on 8 NVIDIA GH200 GPUs (each with ~80 GB memory) for approximately 6 hours. We also experimented with training our model on a single A6000 GPU (48 GB memory). While this setup required more time (approximately 90 hours), it achieved comparable performance (PSNR on RE10K: 25.018 with A6000 vs. 25.033 with GH200). For the *512×512* version, training was performed on 16 NVIDIA GH200 GPUs and required approximately one day.
>
> These details have been added to the updated draft in Appendix A.
>
> ---
> > **Q11: Appendix Tab. 6 ablation does not provide the complete picture. The NVS metrics are dependent on the accuracy of the 3DGS field. Since the 3DGS parameter head is randomly initialized irrespective of the backbone, its contribution due to backbone initialization might not be as pronounced. A more accurate ablation would be on pose estimation given pure 3D points.**
>
> Thank you for your insightful suggestion. To provide a more complete picture, we have added results for pose estimation on RE10K using different weight initialization methods, as shown in the table below. Following your recommendation, we conducted this experiment by estimating poses using pure 3D points (via PnP) without relying on the 3DGS parameters.
>
> The results demonstrate that using CroCov2 as the backbone initialization achieves promising performance, albeit slightly lower than MASt3R and DUSt3R.
> | Init Weights |   5°  |   10° |   20° |
> |---------------|:-----:|:-----:|:-----:|
> | MASt3R      | 0.572 | 0.728 | 0.833 |
> | DUSt3R      | 0.570 | 0.729 | 0.833 |
> | CroCov2     | 0.503 | 0.680 | 0.802 |
>
> ---
> > **Typos and writing suggestions**
>
> Thank you for pointing out these typos and writing suggestions. We have carefully reviewed and addressed them in the updated manuscript.

---

> > ### Comment · Reviewer_wkJy · 2024-11-24
> >
> > I appreciate the authors' detailed responses. The rebuttal clarified my concerns. I have updated my scores accordingly.
> >
> > To summarize, this work builds on top Mast3r to predict 3DGS-parameterized scenes in a common coordinate frame given a pair of input images. In addition, this method shows flexibility in incorporating more than 2 views although this is not the focus of their work. This work differs in two important aspects compared to the closest prior work Splatt3r - 1) they show that finetuning the Mast3r backbone is beneficial for performance rather than freezing it like in Splatt3r, 2) supervising on photometric losses can improve underlying geometry and view synthesis, allowing it to scale to more datasets unlike Splatt3r which requires depth information.
> >
> > I would suggest the authors to remove NVS comparisons on Dust3r and Mast3r since the eval task does not align with the contributions of the original papers.

---

> > > ### Author Response · Authors · 2024-11-24
> > >
> > > Thank you for your response! We’re delighted to hear that our reply addressed your concerns and appreciate your recognition of our work, as well as your summary of the differences compared to the concurrent work, Splatt3R. We truly appreciate your thorough and valuable feedback, which has greatly helped improve our work. Following your suggestion, we will remove the NVS comparisons on Dust3r and Mast3r in the final version.

---

### Official Review · Reviewer_hSJP · 2024-11-02

**Soundness:** 4
**Presentation:** 4
**Contribution:** 4
**Rating:** 8
**Confidence:** 4

**Summary:**

This submission proposes a sparse Gaussian splats prediction given very sparse unposed images. Comparing with prior works such as Master and Splatter, the method achieves good results without using supervision from depth, loss masks, or metric poses.

As this is a strong submission, the only weakness I could think of is that the paper should include quantitative comparisons on geometry reconstruction results with baselines such as Master and Splatter.

In short, my initial recommandation is **accept**, and I am willing to increase it to strong accept if the quantitative comparisons on geometry reconstruction are provided.

**Strengths:**

* The paper is very well written and easy to follow.
* Good reconstruction and NVS results while no need for depth, loss masks, and metric poses comparing with prior work.
* Good generalization capability on in-the-wild data.

**Weaknesses:**

It would be great to include quantitative results for geometry reconstruction, comparing it with Master and Splatter. Currenly only qualitative results are provided in Fig. 5.

**Questions:**

1. Add quantitative results for geometry reconstruction?
2. Clarifying questions (not affecting the paper rating):
    1. It seems like the main difference between this and Splatter is using / not using depth supervision and metric poses, which leads to an "impression" that using depth supervision would limit the dataset choices and hence constrain the GS prediction performance?
    2. L838-840: "Using the ground truth intrinsic during training also causes failure". I assume this is about training Splatter failed using the ground truth intrinsics?
    3. What is the training hardware and training time?
    4. The proposed method also loaded pre-trained Master weights, are they frozen or fine-tuned?
3. Related work: maybe worth adding another concurrent work: ReconX (not affecting the paper rating)

---

> ### Author Response · Authors · 2024-11-23
> **Official Response by Authors -- Part 1**
>
> We would like to sincerely thank the reviewer for the detailed and constructive comments on our work. We take every comment seriously and hope our response can address the reviewer’s concerns. If there are any remaining questions, we are more than happy to address them.
>
> > **Q1: Add quantitative results for geometry reconstruction with MASt3R and Splatt3R?**
>
> Thank you for your valuable suggestion. We would like to clarify first that MASt3R focuses primarily on point-matching quality and pose estimation, while Splatt3R evaluates performance only through novel view synthesis experiments. Additionally, the datasets we use for evaluation in the paper do not include ground truth (GT) geometry annotations, which limits the ability to perform direct quantitative evaluations.
>
> Despite these challenges, **we have included a qualitative comparison of geometry reconstruction by visualizing 3D Gaussians or point clouds** in our revised manuscript, which can be found in Fig. 12 and L.955–967. The results demonstrate that our approach captures finer geometric details and higher geometry quality than both MASt3R and Splatt3R.
>
> This outcome is particularly encouraging, as our method obtains such high-quality geometry using only photometric loss as supervision during training, while both MASt3R and Splatt3R rely on ground-truth point cloud supervision.
>
> We hope this comparison addresses your concern effectively.
>
> ---
> > **Q2: It seems like the main difference between this and Splatter is using / not using depth supervision and metric poses, which leads to an "impression" that using depth supervision would limit the dataset choices and hence constrain the GS prediction performance?**
>
> Thanks for your observation! Below we highlight the main differences to Splatt3R:
>
> 1. *Dataset Flexibility and Depth Supervision*
> We agree with the reviewer that one main difference between our approach and Splatt3R is their reliance on depth/point cloud supervision, which inherently limits the range of datasets that can be utilized for training. In contrast, our method does not require depth supervision or metric poses during training. This flexibility enables training with a broader range of commonly available video datasets, such as RE10K and DL3DV, ultimately improving our performance.
> 2. *End-to-End Training for Enhanced Performance*
> Another key difference is in training strategy. Unlike Splatt3R, which freezes the pretrained weights of MASt3R and trains only the Gaussian head, our method trains all network parameters end-to-end. This design choice addresses potential misalignments in the point clouds predicted by MASt3R, leading to more consistent renderings (see Fig. 4 in the paper). Additionally, this strategy supports varying numbers of input views (e.g., three views, as shown in Tab. 5), further demonstrating its robustness.
> 3. *RGB Image Shortcut for Improved Detail Capture*
> We also incorporate an RGB image shortcut in the Gaussian head (L.224–227), which allows our method to better capture fine texture details for Gaussian prediction, see row (e) of Tab. 5.
> 4. *Intrinsic Injection for Scale Ambiguity Resolution*
> As detailed in Sec. 3.4, we also include an intrinsic injection module to explicitly address the scale ambiguity problem and further boost performance.
>
> We will update our paper to incorporate the comparison with Splatt3R, highlighting these distinctions and the resulting improvements in performance.
>
> ---
> > **Q3: L838-840: "Using the ground truth intrinsic during training also causes failure". I assume this is about training Splatter failed using the ground truth intrinsics?**
>
> Thank you for carefully reviewing our paper and bringing up this point.
>
> The failure of Splatt3R when using ground truth (GT) intrinsics during training occurs because they freeze the pretrained weights of MASt3R. As a result, the GT intrinsics do not align with the scale of the Gaussian centers predicted by MASt3R. This misalignment leads to rendered images that are inconsistent with the ground truth images during training, introducing noisy supervision signals and ultimately causing the training process to fail.
>
> ---
> > **Q4: What is the training hardware and training time?**
>
> For the *256×256* version of the model, training was conducted on 8 NVIDIA GH200 GPUs (each with ~80 GB memory) for approximately 6 hours. We also experimented with training our model on a single A6000 GPU (48 GB memory). While this setup required more time (approximately 90 hours), it achieved comparable performance (PSNR on RE10K: 25.018 with A6000 vs. 25.033 with GH200). For the *512×512* version, training was performed on 16 NVIDIA GH200 GPUs and required approximately one day.
>
> These details have been added to the updated draft in Appendix A.

---

> ### Author Response · Authors · 2024-11-23
> **Official Response by Authors -- Part 2**
>
> > **Q5: The proposed method also loaded pre-trained Master weights, are they frozen or fine-tuned?**
>
> The loaded pretrained MASt3R weights are fine-tuned as part of our end-to-end training strategy, leading to improved performance and consistent renderings.
>
> ---
> > **Q6: Related work: maybe worth adding another concurrent work: ReconX**
>
> Thanks for your suggestion. The main difference is that our approach employs a feed-forward reconstruction method, while ReconX requires per-scene optimization and is, therefore, less efficient. We have incorporated a reference to ReconX and included an analysis in our revised manuscript. Please see L.180-L.182 for the updated details.

---

> ### Comment · Reviewer_hSJP · 2024-11-25
> **Thanks for the detailed response**
>
> Dear Authors,
>
> Thank you for your detailed responses. I would like to maintain my original rating, as I still believe a quantitative comparison of geometry reconstruction is feasible—for instance, using a dataset with ground truth depth. However, I understand that implementing such an evaluation may not be practical within the rebuttal period.
>
> Best regards,
> Reviewer hSJP

---

> > ### Author Response · Authors · 2024-11-28
> >
> > Dear Reviewer hSJP,
> >
> > Thank you for your thoughtful feedback and recognition of our work. We strongly agree that the quality of geometric reconstruction is an important task. However, as noted in the manuscript, the primary focus of our paper is on novel view synthesis and pose estimation tasks, rather than explicit geometric reconstruction. Moreover, a key consideration is that the 3D Gaussian Splatting (3DGS) representation is inherently optimized for NVS, and as such, the predicted Gaussians are not constrained to lie on the surface. To emphasize geometry quality explicitly, one might explore a reformulated Gaussian representation, such as 2D Gaussian Splatting (2DGS) [1], specifically tailored for accurate geometric reconstruction. This is an exciting direction, and we aim to pursue it in future work.
> >
> > [1] Huang, Binbin, et al. "2d Gaussian splatting for geometrically accurate radiance fields." ACM SIGGRAPH 2024 Conference Papers. 2024.
> >
> > Best regards, Authors.

---

### Author Response · Authors · 2024-11-23
**General Response**

We thank all reviewers for their detailed reviews and suggestions!

We have updated the manuscript with the following revisions based on the reviewers' suggestions. Some experimental results were only included in the responses to the reviewers, but we will include them in the paper later. All revisions in the updated version are highlighted in red：

1. **Add more experiments**:
    - Add qualitative geometry comparison with MASt3R and Splatt3R (hSJP-Q1; L.955).
    - Add experimental results with more input views (wkJy-Q4, tg6H-Q6; L.939, L.946).
    - Add performance comparison on object-level data (tg6H-Q3; Appendix B)
    - Add experimental results using DINOv2 pre-trained weights and random initialization (tg6H-Q3; L.889)
    - Add an ablation study with Plücker ray pose embeddings (tg6H-Q3; L.906)

2. **Add more discussion and details**:
    - Expand related works in Sec. 2: Discuss differences with LEAP, PF-LRM, and other related methods (tg6H-Q1) as well as the concurrent work ReconX (hSJP-Q3).
    - Add training time and hardware requirements (hSJP-Q2, wkJy-Q9; Appendix A).
    - Provide a discussion comparing our approach with previous intrinsic embedding methods (tg6H-Q5; Sec. 3.4).

Thanks again for all the effort and time, and we look forward to further discussions if there are any more questions.

---

### Meta-Review · Area_Chair_k4Xq · 2024-12-19

**Metareview:**

This paper introduces a method for reconstructing 3DGS from sparse and unposed images using only photometric constraints for training. The technique is then used for NVS and pose estimation.

The main strengths of this paper are as below:
- The paper is clearly written and easy to follow.
- The paper is innovative, provides a detailed discussion of previous work, and is well-motivated.
- The paper has presented how to train the 3DGS Reconstruction Network only with photometric constraints and without the geometric ground truth, which enables the utilization of a wider range of datasets and improves the reconstruction performance.
- The paper provides comprehensive experiments that fully demonstrate the sophistication of the proposed method.

Needing more experiments and analyses were raised by the reviewers. Please revise the paper according to the discussions before submitting the final version.

**Additional Comments On Reviewer Discussion:**

Reviewer hSJP primarily asked questions about the implementation details and suggested adding quantitative geometric comparisons. The authors explained the implementation details and added a qualitative geometric comparison (no quantitative comparison).

Reviewer wkJy was concerned about the essential differences with the concurrent work, and suggested adding more evaluation experiments, modifying unreasonable comparisons, and adding more explanations for the experimental results. The authors assuaged the reviewer's concerns and improved the experiments according to the reviewer's suggestions.

Reviewer tg6H questioned the innovativeness, pointing out the lack of discussion of a large body of related work and the irrationality of the experimental design. The authors discussed the related work noted by the reviewer. They conducted the experiments according to the reviewer's requirements and achieved good results.

---

### Decision · Program_Chairs · 2025-01-22

Accept (Oral)